

# Lagrangian-Eulerian statistics of mesoscale ocean chlorophyll from Bio-Argo floats and satellites

Darren C. McKee[1], Scott C. Doney[1], Alice Della Penna[2,3], Emmanuel S. Boss[4], Peter Gaube[5], Michael J. Behrenfeld[6], David M. Glover[7]

[1]Department of Environmental Sciences, University of Virginia, Charlottesville, VA, 22904, USA
[2]Institute of Marine Science, University of Auckland, Auckland, New Zealand
[3]School of Biological Sciences, University of Auckland, Auckland, New Zealand
[4]School of Marine Sciences, University of Maine, Orono, ME, USA
[5]Applied Physics Laboratory, University of Washington, Seattle, WA, USA
[6]Department of Botany and Plant Pathology, Oregon State University, Corvallis, OR, USA
[7]Department of Marine Chemistry and Geochemistry, Woods Hole Oceanographic Institution, Woods Hole, MA, USA

*Correspondence to*: Darren C. McKee (dcm2xp@virginia.edu)

**Abstract.** Phytoplankton form the base of marine food webs and play an important role in carbon cycling, making it important to quantify rates of biomass accumulation and loss. Since phytoplankton drift with ocean currents, rates should be
evaluated in a Lagrangian as opposed to Eulerian framework. In this study, we quantify the Lagrangian (from Bio-Argo floats and surface drifters with satellite ocean colour) and Eulerian (from satellite ocean colour and altimetry) statistics of mesoscale chlorophyll and velocity by computing decorrelation time and length scales and relate the frames by scaling the material derivative of chlorophyll. Because floats profile vertically and are not perfect Lagrangian observers, we quantify the mean distance between float and surface geostrophic trajectories over the time spanned by three consecutive profiles (Quasi-
Planktonic Index; QPI) to assess how their sampling is a function of their deviations from surface motion. Lagrangian-Eulerian statistics of chlorophyll are sensitive to the filtering used to compute anomalies. Chlorophyll anomalies about a 31-day time filter reveal approximate equivalence of Lagrangian and Eulerian tendencies, suggesting they are driven by ocean-colour-pixel-scale processes and sources or sinks. Chlorophyll anomalies about a seasonal cycle have Eulerian scales similar to those of velocity, suggesting mesoscale stirring helps set distributions of biological properties, and ratios of Lagrangian to
Eulerian timescales depend on observer speed relative to an evolution speed of the chlorophyll fields in a manner similar to earlier theoretical results for velocity scales. By lagging surface chlorophyll patches, floats underestimate the Lagrangian tendency and advective terms, and the Eulerian tendency primarily sets timescales; however, since the QPI increases with profiling interval, frequent profiling can generate more accurate time series of phytoplankton accumulation.

## 1 Introduction

Upper-ocean phytoplankton communities vary on sub-diurnal and sub-seasonal timescales and submesoscale to mesoscale spatial scales. Fully capturing this variability is challenging because of the temporal and spatial limitations of different





observational platforms, choices associated with sampling strategies, and data gaps, creating the need to best leverage a variety of complementary observing platforms (Chai et al., 2020). Time derivatives of surface chlorophyll and phytoplankton carbon provide valuable estimates of the phytoplankton net specific accumulation rate ($r$) that reflect biological growth and

loss processes as well as physical advection and mixing (e.g., Behrenfeld et al., 2005). The temporal variability of $r$ from Eulerian time series from, for example, a mooring or high-resolution ship observations at a fixed geographic location necessarily incorporates a variance component from advective and mixing divergence. Similar issues arise in the analysis of $r$ from satellite ocean colour data on fixed geographic grids, with the additional complication of temporal data gaps caused by satellite orbital dynamics and cloud cover.

In principle, a Lagrangian or water-parcel following framework isolates net biological growth from horizontal physical transport, allowing more direct comparisons to laboratory and mesocosm biological experiments, theory, and food-web models. Analysis of many Lagrangian series reveals sensitivity of phytoplankton community growth rates to environmental conditions experienced (Zaiss et al., 2021), allows for partitioning of chlorophyll (Chl) variance into net community production and advective effects (Jönsson et al., 2011), and reveals how dispersion regulates phytoplankton blooming

(Lehahn et al., 2017). Records from surface or mixed-layer drifters with bio-optical sensors are rare and often of short duration (Abbott and Letelier, 1998; Briggs et al., 2018). Alternatively, one can obtain Lagrangian time series by projecting satellite ocean colour data onto surface trajectories (Jönsson et al., 2009), either those from in situ surface drifters or from synthetic particles advected with surface currents from ocean models or satellite altimetry. This approach has yielded important insights into the roles of episodic events in controlling net community production in coastal regions (Jönsson and

Salisbury, 2016) and of submesoscale biophysical dynamics at ocean fronts (Zhang et al., 2019). Nevertheless, it ultimately falls victim to the limited spatial information content of any ocean colour product (Doney et al., 2003; Glover et al., 2018).

An alternative, complementary observing strategy involves Bio-Argo floats, a platform experiencing a rapid growth in deployments for monitoring ocean biogeochemistry and ecosystems (Claustre et al., 2010; Gruber et al., 2010; https://biogeochemical-argo.org/). Bio-Argo floats are like traditional Argo floats but equipped with additional sensors to

measure variables such as chlorophyll fluorescence, backscatter, and/or nutrient concentrations. Depth resolution of these variables in combination with hydrographic variables allows floats to detect rare or small-scale events, such as wintertime restratification by mixed layer instabilities (Lacour et al., 2017), subduction of particulate organic carbon (Llort et al., 2018), and upwelling due to rapid evolution of mesoscale eddies (Ascani et al., 2013). However, formally, Bio-Argo floats are only quasi-Lagrangian, reflecting a weighted average of velocities experienced between their parking depth and the surface. To

properly sample evolution of ocean mixed layer biology, a platform should be nearly Lagrangian with respect to the surface flow. Typically, floats profile every few days, meaning they spend most of their time drifting with more sluggish flows at a parking depth of ~1,000 m. However, when vertical shear is weak or the floats profile more frequently, Bio-Argo floats might serve as a viable platform for studying evolution of upper-ocean phytoplankton communities.

In this paper we seek to understand the Lagrangian statistics (time and length scales) of mesoscale Chl anomalies in a

subregion of the North Atlantic Ocean and how these depend on the underlying Eulerian statistics of the Chl field and a





water parcel's motion. Because Chl is stirred, we first diagnose the Lagrangian-Eulerian statistics of the velocity field. We take a trajectory-scale perspective, drawing on earlier theoretical (Middleton, 1985) and observational (Lumpkin et al., 2002) studies using the framework of the material derivative to quantify the relative contributions of advective and tendency terms. In particular, we characterize the fields by computing integral time and space scales of autocorrelation functions from floats,

surface drifters, and satellite altimetry fields. We also take a local perspective by constructing a Quasi-Planktonic Index (QPI; Della Penna et al., 2015) that quantifies the distance between a float trajectory and synthetic surface trajectories (from altimetric geostrophic currents) over three consecutive profiles. We combined these two perspectives to highlight quasi-Lagrangian behaviour of floats (that affects sampling) by weighting their averaged integral time scales by the inverse-squared median QPI over individual time segments. Similar to the velocity analysis, we compute integral time scales of Chl

for floats, ocean colour projected onto surface drifter tracks, and Eulerian fixed-location pixels of ocean colour, and evaluate them through the framework of the material derivative. Scales of Chl and velocity are compared to assess correspondence. Because data sparsity (long profiling interval of floats and gaps in ocean colour) influences our autocorrelation functions, we also evaluate decorrelation times from the power spectra of the Lagrangian series, providing a complementary perspective. Our analysis builds on recent regional studies of the spatial geostatistics of satellite ocean colour (Eveleth et al., 2021;

Glover et al., 2018) and the seasonal to annual variations in phytoplankton chlorophyll, carbon biomass, and net primary production from Bio-Argo floats (Yang, 2021; Yang et al., 2020).

## 2 Framework

### 2.1 Material derivative and integral scales

The material derivative of a scalar such as $Chl(\mathbf{x}(t), t)$ is

$$\underbrace{\frac{D Chl}{Dt}}_{\text{Lagrangian tendency}} = \underbrace{\frac{\partial Chl}{\partial t}}_{\text{Eulerian tendency}} + \underbrace{\mathbf{u} \cdot \nabla Chl}_{\text{Advection}} = S \tag{1}$$

where $S(\mathbf{x}(t), t)$ represents sources and sinks along trajectory $\mathbf{x}(t)$, and $d\mathbf{x}/dt = \mathbf{u}$. If Chl were conserved, $S = 0$. A scalar or velocity field exhibits decorrelation in space and time. From an Eulerian perspective, decorrelations of velocity can be quantified with integral scales $T_E$ and $L_E$. A Lagrangian sampling platform moving with the surface flow will experience spatial and temporal velocity decorrelations simultaneously, mixing the field's temporal and spatial information, and so will

tend to exhibit a shorter decorrelation time compared to an Eulerian observer ($T_L \leq T_E$). Assuming the variances of Chl are equal in each frame, the ratio of the advective and tendency terms in Eq. (1) scales as:

$$\frac{u'/L_E}{1/T_E} = \frac{u'}{L_E/T_E} = \frac{u'}{c^*} = \alpha \tag{2}$$

where $u'$ is a scale for the mesoscale eddy velocities, $L_E$ and $T_E$ are Eulerian length and time scales for the mesoscale velocity field, and $c^* = L_E/T_E$ is an evolution speed for the eddy field.



Philip (1967) argued that the quantity $T_L / T_E$, a measure of the difference in Lagrangian and Eulerian perspectives, should depend only on α. For a homogenous and stationary 2-D eddy field, Middleton (1985) assumed certain functional forms for the Eulerian energy spectrum and assumed the distribution of parcel displacements was stationary and Gaussian to determine the relations

$$T_L/T_E = q\left(q^2 + \alpha^2\right)^{-1/2} \tag{3a}$$

$$L_L/L_E = \alpha q\left(q^2 + \alpha^2\right)^{-1/2} \tag{3b}$$

where $q = \sqrt{\pi/8}$. To interpret their meaning, consider the case where $\alpha \ll 1$. In this case, the tendency term dominates the advective term, or equivalently, the platform is advected more slowly than the eddy field evolves and the velocity decorrelation is determined by Eulerian temporal evolution ($T_L \approx T_E$). This renders the platform like a mooring, and this regime is referred to as the "fixed-float" regime (terminology as in LaCasce, 2008). On the other hand, suppose that $\alpha \gg 1$. In that case, the advective term dominates, or equivalently, the platform is advected across eddies faster than they evolve and the Lagrangian decorrelation of velocity is determined by the temporal imprint of spatial decorrelations ($T_L < T_E$). This is referred to as the "frozen-turbulence" regime (as in LaCasce, 2008), related to Taylor's hypothesis (Taylor, 1938).

Lumpkin et al. (2002) applied Eq. (3) to surface drifters and deep isopycnal floats, computing Lagrangian integral time scales from those platforms and computing Eulerian integral time scales from an ocean model. They found the theoretical model to hold well: the deep floats fell in the "fixed-float" regime and the surface drifters spanned the two regimes, with spatial variability accounted for by variability in the kinetic energy of major current systems in the North Atlantic. Since Bio-Argo floats profile, it is not clear what regime they should experience. Our first step is to compute Lagrangian velocity scales ($T_L$, $L_L$) from trajectories of the Bio-Argo floats and drifters and to evaluate how the floats move horizontally by evaluating the relations in Eq. (3), where Eulerian velocity scales ($T_E$, $L_E$) are calculated from maps of surface geostrophic velocity anomalies from satellite altimetry. If the flow is dominated by mesoscale balanced motions, flows at parking depth should mimic those at the surface with a reduction in magnitude (which does not affect decorrelation time) and a slight decay of high wavenumbers (Klein et al., 2009; Lapeyre and Klein, 2006), allowing geostrophic Eulerian scales to be compared to both drifters and floats.

Such an analysis yields a statistical representation of how an observer moves horizontally but does not directly inform us of the statistics of Chl that are sampled by the moving platform. To do that, we analyze a scaling of the material derivative using time and length scales of Chl rather than velocity. While there is not a theoretical relationship equivalent to Eq. (3) for tracers and no a priori relation between $T_{L,\text{Chl}}$ and $T_{E,\text{Chl}}$, we first explore the equivalent parameter spaces

$$T_{L,\text{Chl}}/T_{E,\text{Chl}} = F\left(\alpha_{\text{Chl}}\right) \tag{4a}$$

$$L_{L,\text{Chl}}/L_{E,\text{Chl}} = G\left(\alpha_{\text{Chl}}\right) = \alpha_{\text{Chl}} F\left(\alpha_{\text{Chl}}\right) \tag{4b}$$





where $\alpha_{Chl} = u'/c_{Chl}^*$ and $c_{Chl}^* = L_{E,Chl}/T_{E,Chl}$. We envision Eq. (4) as a parallel to Eq. (3) with equivalent interpretation, namely a quantification of how Lagrangian and Eulerian time and length scales of Chl vary as a function of how an observer (float or drifter) traverses evolving space-time Chl fields. Additionally, we then admit a scalar variance that may vary by reference frame to obtain the scaling

$$\frac{\langle \text{Chl} \rangle_L}{T_{L,Chl}} = \frac{\langle \text{Chl} \rangle_E}{T_{E,Chl}} + \langle u' \rangle \frac{\langle \text{Chl} \rangle_{space}}{L_{E,Chl}} . \tag{5}$$

With this more general approach we assess the relative magnitude of the three scaling terms in Eq. (5), which are from left to right the Lagrangian tendency (LAG), Eulerian tendency (EUL), and eddy advection of mesoscale gradients (ADV). The Eulerian scales are derived from satellite ocean colour, and the Lagrangian scales $\langle \text{Chl} \rangle_L$, $T_{L,Chl}$, and $\langle u' \rangle$ are derived from floats (Chl measured by onboard fluorometer) and drifters (Chl from satellite ocean colour projected onto trajectories). We expect the relative magnitudes of the Lagrangian and advective terms to be different between floats and drifters. For both the

velocity and Chl analyses, all necessary scales and the datasets used to estimate them are summarized in Table 1. The methodology (integral of autocorrelation function) and datasets are described in Sect. 3.

| Scale | Definition | Source | Time window | ACF bin |
|---|---|---|---|---|
| $T_L$ | Lagrangian velocity timescale | Floats<br>Drifters | 120 day<br>120 day | 5 day<br>1 day |
| $T_E$ | Eulerian velocity timescale | Altimetry | 120 day | 1 day |
| $L_E$ | Eulerian velocity length scale | Altimetry | N/A | 27.8 km |
| $T_{L,Chl}$ | Lagrangian Chl timescale | Floats<br>Drifters w/projected ocean colour<br>metbio003d segment<br>metbio010d segment | 120 day<br>120 day<br>55 day<br>48 day | 5 day<br>1 day<br>1 day<br>1 day |
| $T_{E,Chl}$ | Eulerian Chl timescale | GlobColour pixel time series | 365-366 day | 1 day |
| $L_{E,Chl}$ | Eulerian Chl length scale | Glover et al. (2018) variogram ranges | N/A | N/A |
| $u'$ | Lagrangian velocity scale | Floats<br>Drifters | 120 day<br>120 day | N/A |
| $\langle \text{Chl} \rangle_L$ | Lagrangian Chl scale | Floats<br>Drifters w/projected ocean colour<br>metbio003d segment<br>metbio010d segment | 120 day<br>120 day<br>55 day<br>48 day | N/A |
| $\langle \text{Chl} \rangle_E$ | Eulerian Chl scale | GlobColour pixel time series | 365-366 day | N/A |
| $\langle \text{Chl} \rangle_{space}$ | Spatial Chl scale | Glover et al. (2018) variogram relative sills | N/A | N/A |

**Table 1:** Overview of the time and space scales and variances, the data sources from which they are calculated, and the bin sizes used for the discrete temporal or spatial autocorrelation functions (ACFs). All time scales and time variances are computed from non-overlapping
segments. Length scales for velocity are derived from discrete radial (isotropic) ACFs in 5° x 5° space bins. Length scales for chlorophyll (Chl) are derived from the variograms calculated by Glover et al. (2018).





### 2.2 Quasi-Planktonic Index (QPI)

Float velocities are estimated by centered differencing positions of neighbouring profiles. While the preceding material derivative analysis provides a holistic summary of how a float samples mesoscale fields over some time window, it is also
useful to obtain a more local measure of the similarity of float and Lagrangian trajectories. We construct a Quasi-Planktonic Index (QPI) that quantifies the similarity of the float trajectory to a best-fit synthetic surface trajectory advected by altimetric total geostrophic currents. This index is similar to the one developed by Della Penna et al. (2015) but is tailored to evaluate the centered difference derivatives. At each time step $t_i$ of a float trajectory, we advect a disk of particles of radius 0.3° both forwards (to $t_{i+1}$) and backwards (to $t_{i-1}$) in time. For each synthetic trajectory we compute the distance between itself and the
true float trajectory and choose the trajectory that minimizes the average distance over the three time steps ($t_{i-1}$, $t_i$, $t_{i+1}$), with the average distance being the QPI (in kilometres). Full details of the calculation are given in Appendix A. To tie the two frameworks together, we hypothesize that a Bio-Argo float with a smaller median QPI over some period of time will behave more like a surface drifter, with a larger $\alpha$ and a smaller $T_L / T_E$.

### 3 Data and methods

### 3.1 Study region

Our study domain approximately corresponds to that of the North Atlantic Aerosols and Marine Ecosystems Study (NAAMES) field campaign in the subtropical to subpolar transition region of the North Atlantic Ocean (Behrenfeld et al., 2019). The domain boundaries were chosen to encompass the full trajectories of the Bio-Argo floats that we analyze. The domain includes the typical spatial extent of the North Atlantic spring bloom and includes the high-strain and high-eddy
kinetic energy conditions of the North Atlantic Current sandwiched between more quiescent subpolar and subtropical conditions. We tile the domain into 5° x 5° cells as done by Glover et al. (2018) (Figure 1) and compute averaged integral scales in each, using satellite pixels or float or drifter segments whose median latitude and longitude reside within.





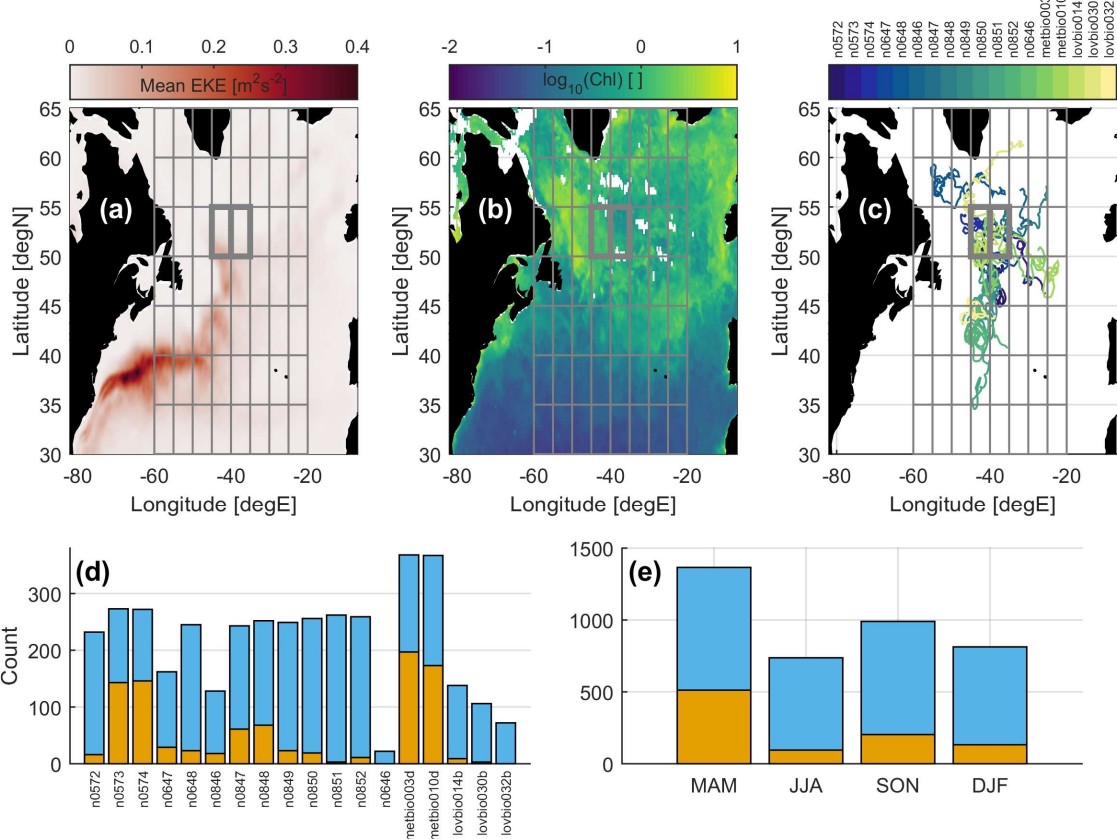

**Figure 1:** Overview of study domain and float sampling. **(a)** Time-mean geostrophic eddy kinetic energy from altimetry; **(b)** Snapshot of
$\log_{10}$(Chl) to convey a typical bloom (June 2002; from GlobColour); **(c)** Locations of all float tracks; **(d)** Counts of all profiles (blue) and
profiles with Quasi-Planktonic Index (QPI) < 5 km (orange); **(e)** As in (d) but profiles are binned according to season. In panels (a)-(c) we
display the 5º x 5º space bins used for computing Lagrangian and Eulerian scales. The two bolded bins host the rapidly profiling metbio*
float segments that are the subject of further analysis.

### 3.2 Data

#### 3.2.1 Floats

We use data from 13 Bio-Argo floats deployed over four cruises during NAAMES (Figure 1; floats with prefix 'n'). Most of
the floats are confined to the high-strain and high-eddy kinetic energy conditions of the North Atlantic Current (Figure 1a).
In addition to measuring salinity, temperature, and pressure like standard Argo floats, the NAAMES floats measured
backscatter at 700 nm and chlorophyll fluorescence. Chlorophyll a concentration (Chl) is derived from fluorescence and is
calibrated against discrete HPLC samples and corrected for non-photochemical quenching (Xing et al., 2012). All float data


are obtained as L2 data from the University of Maine In-Situ Sound and Color Lab web archive and are distributed as profiles with 2-m resolution in the vertical between 0-500 m (4 m between 500-1,000 m). All profile quantities are interpolated to a 1-m grid using a cubic hermite interpolating polynomial. The Chl data are quality controlled by the U. Maine group. For temperature and salinity, when possible we match the L2 files to profile files in the Argo GDAC and keep

only samples with adjusted profiles with a QC flag of 1, 2, 3, 5, or 9 prior to interpolating (flags 2 and 3 are omitted for any "Real Time" profiles).

We include 5 additional floats that were used to inform NAAMES station sampling but were deployed by other projects (Figure 1; lovbio* and metbio* floats). Data were obtained as Sprof files from the Argo GDAC and only profiles overlapping in time with NAAMES were retained. While "Delayed Time" hydrographic profiles were generally available,

only "Adjusted" Chl profiles were available, meaning only an automated set of quality checks have been applied (Schmechtig et al., 2018). Samples with QC flags of 1, 2, 3, 5, and 9 are retained and the profiles are interpolated as with the NAAMES floats. Throughout, special attention is paid to brief (~50 day) segments from metbio003d and metbio010d that exhibited consistently frequent and shallow profiling as they may exhibit more closely Lagrangian behaviour.

For all floats the mixed layer depth (MLD) is computed as the depth at which the potential density exceeds that at 10 m by

0.03 kg m$^{-3}$ and we construct a single time series of Chl for each float by averaging over the MLD and taking the base 10 logarithm. To remove sub-daily variability, Chl time series are smoothed with a running 48-hour Hamming window. Float velocities are estimated by centered differencing profile positions and the QPI is calculated for each profile as in Appendix A.

### 3.2.2 Satellite data

We use the multi-satellite merged altimetry dataset distributed by Copernicus Marine Environmental Services which includes daily maps of surface geostrophic velocity and geostrophic velocity anomalies on a 0.25º grid. Geostrophic velocities are used for particle advection when computing the QPI and for comparing to effective float or drifter velocities. The geostrophic velocity anomalies (from sea level anomalies relative to a long-term mean) effectively isolate the mesoscale and are used for computing scales $T_E$ and $L_E$. A high pass filter is applied to each anomaly time series by Fourier

Transforming and zeroing out frequencies lower than (150 days)$^{-1}$ before inverse Fourier Transforming back to the time domain.

We obtain fields of log-transformed, daily, 0.25º, L3m Chl fields from GlobColour computed from the GSM algorithm and blending all available satellites. We use a relatively spatially coarse and blended product to maximize data coverage. All maps are assumed to correspond to 12:00:00 UTC. The time domain for all satellite data used to calculate integral scales is

January 2003 – December 2016 and approximately follows the Glover et al. (2018) study.





### 3.2.3 Drifters

Six-hourly surface drifter trajectories within the time domain of the satellite data were obtained from the NOAA Global Drifter Program. The dataset reports velocities obtained by centered differencing positions. To remove the influence of inertial oscillations and tides, which can decrease $T_L$ compared to fluctuations due to mesoscale processes, we filter every

drifter velocity series with an ideal low pass filter by zeroing out all frequencies lower than 2/(3 IP) in the frequency domain, where IP is the inertial period corresponding to the trajectory's median latitude. For series with gaps, the filter is applied to individual segments so long as they are longer than 20 days. Finally, the filtered time series are subsampled once per day: at 00:00:00 UTC for comparison with the altimetry fields or at 12:00:00 UTC for comparison with the ocean colour fields. We construct Lagrangian time series of Chl by bilinearly interpolating the daily mesoscale Chl maps onto the subsampled drifter

returns.

### 3.3 Subtrahends for chlorophyll

A subtrahend is a field to be subtracted from another. To isolate mesoscale Chl variability, all Chl integral scales are computed from anomalies relative to one of two Chl subtrahends that we construct. More details of the methodology and an example time series are given in Appendix B.

The first subtrahend ("smoothed") is meant to replicate that used by Glover et al. (2018) so that we can obtain Lagrangian and Eulerian time scales consistent with their Eulerian space scales. All GlobColour space-time fields are convolved with a 3-D filter defined as a 31-day Hamming window in time and a 2-D Gaussian in space with a 1º full-width-half-maximum and a 2º cutoff. For drifters, the anomalies are projected onto the drifter tracks. For floats, we perform a weighted running average of each float Chl series with weights defined by a 31-day Hamming window. The objective of the space-time

filtering in Glover et al. (2018) was to isolate mesoscale (and any resolved submesoscale) variability signals (anomalies) from the lower frequency seasonal and geographic patterns of bloom formation and decline that were meant to be captured in the subtrahend. However, the short 31-day time window may have the undesirable effect of retaining some of the mesoscale signal in the subtrahend, instead isolating faster processes in Chl anomalies. Yang et al. (2020) found that phytoplankton accumulation rates $r$ are typically one to two orders of magnitude smaller than growth rates so that intraseasonal $r^{-1}$ is

generally $O(10$ days$)$. Similar physical time scales are found for lifetimes (weeks; Chelton et al., 2011; Gaube et al., 2014) and inverse growth rates (days to weeks; Smith, 2007) of energy-containing mesoscale eddies.

Given those concerns, the second subtrahend is meant to strictly isolate anomalies from a repeating annual cycle ("climatology"). All GlobColour space-time fields are convolved with a 4-D filter defined as a 31-day-of-year Hamming window in time, a 2-D Gaussian in space with a 1º full-width-half-maximum and a 2º cutoff, and a boxcar function (equal

weights) across years. For drifters, the anomalies are projected onto the drifter tracks. For floats, the subtrahend is projected onto the float tracks, regressed against float data to account for different data dynamic ranges, and differenced.





### 3.4 Integral time scales

Our goal is to obtain estimates of the Lagrangian and Eulerian integral time scales of velocity and Chl representative over a 5° x 5° bin for each measurement platform (Table 1). For all platforms (floats, drifters, satellite-derived fields), all individual time series of variable $y(t)$ (zonal [$u$] or meridional [$v$] velocity or Chl anomalies) are broken into 120-day segments (365-day for Eulerian ocean colour) and each segment is prewhitened, either by removing the scalar mean (Chl anomaly series) or a linear trend line fit by regression (velocity series), creating $y'(t)$ with zero mean. The latitude-longitude coordinates for each time series segment are defined as the median latitude and longitude over the segment length for drifting platforms (float, drifters) or as the pixel latitude and longitude.

Integral time scales are estimated from autocorrelation functions (ACF). For zero-mean data, the discrete autocovariance function (ACVF) at lag $\tau$ is

$$C(\tau) = \frac{1}{N(\tau)} \sum_{k=1}^{N(\tau)} \left[ y'(t_k) y'(t_k + \tau) \right] \tag{6}$$

where $N(\tau)$ is the number of data points in $y'_i$ separated by $(n-1)\Delta\tau \leq \tau < n\Delta\tau$. To arrive at an ACF, $C(\tau)$ needs to be normalized by the variance, which explicitly is $C(0) = \overline{y'^2_i}$. However, for unevenly spaced (equivalently gapped) data, dividing by $\overline{y'^2_i}$ can lead to ACF values greater than 1. Because $y'_i$ is stationary, $\sqrt{\overline{y'^2_i(t)y'^2_i(t+\tau)}} = \overline{y'^2_i(t)}$ and the normalization issue can be avoided by dividing $C(\tau)$ by a measure of the variance using only the data points that went into the calculation at lag $\tau$,

$$R(\tau) = C(\tau) / J(\tau) \tag{7a}$$

where

$$J(\tau) = \frac{1}{N(\tau)} \left[ \left( \sum_{k=1}^{N(\tau)} y'(t_k)^2 \right) \left( \sum_{k=1}^{N(\tau)} y'(t_k + \tau)^2 \right) \right]^{1/2}. \tag{7b}$$

At this point, we could obtain the integral scale $T$ by integrating $R(\tau)$ to the lag of its first zero crossing, which for a discrete ACF is $T = \sum_{\tau < \tau_0} \left[ R(\tau) \right] \Delta\tau$. Applying this method to all segments $i$ from all time series in some space subset $I$ (e.g., a 5° x 5° bin), we could then obtain average scales $\overline{T}$ by averaging estimates from all $i \in I$. An alternative approach to obtain a spatially averaged scale would be to first construct a single composite ACF that is representative of $I$ and then integrate that single ACF to its first zero crossing. To construct such an ACF, we use pairs of points from all locally prewhitened segments $y'_i(t)$ to construct a composite $R^c(\tau) = C^c(\tau) / J^c(\tau)$ where





$$C^c(\tau) = \frac{\sum_{i \in I}\left[\sum_{k=1}^{N_i(\tau)}\left[y_i{}'(t_k)\,y_i{}'(t_k+\tau)\right]\right]}{\sum_{i \in I} N_i(\tau)} \tag{8a}$$

$$J^c(\tau) = \frac{\sqrt{\sum_{i \in I}\left[\sum_{k=1}^{N_i(\tau)}\left(y_i{}'(t_k)^2\right)\right]\sum_{i \in I}\left[\sum_{k=1}^{N_i(\tau)}\left(y_i{}'(t_k+\tau)^2\right)\right]}}{\sum_{i \in I} N_i(\tau)} \; . \tag{8b}$$

For this composite ACF, $T^c = \sum_{\tau < \tau_0}\left[R^c(\tau)\right]\Delta\tau$. For any dataset involving satellite ocean colour, $T^c$ must be used due to the

large number of gaps. For evenly spaced data sets we find $T^c \approx \overline{T}$, but $T^c$ can be quite different from $\overline{T}$ for the floats with $\overline{T}$ biased larger. This is because $\overline{T}$ weighs each segment $i$ equally while segments with shorter median $\Delta t$ between profiles contribute more to $R^c(\tau)$ than do those with a longer median $\Delta t$ by virtue of offering more data pairs. In this regard, $R^c(\tau)$ is a better estimate of the ensemble ACF and $T^c$ a better estimate of the ensemble integral scale. However, there is value in continuing to use $\overline{T}$ when possible since this reverse order of operations allows us to construct average scales weighted by (QPI)$^{-2}$, allowing us to gauge in parameter space how a smaller median QPI over the window size affects the turbulence regime experienced by the floats.

All time scales analyzed in this study are either derived by averaging in space (from integrating Eq. (7) and averaging) or derived from space-composited ACFs (from integrating Eq. (8)), with each method using all segments in the 5º x 5º space bins depicted in Fig. 1. For Lagrangian velocity time scales, $T_{L,u}$ and $T_{L,v}$ representative of a space bin are calculated from all enclosed float or drifter segments by integrating Eq. (7) and spatially averaging before taking $T_L = 0.5\left(\overline{T_{L,u}} + \overline{T_{L,v}}\right)$. The same is done for Eulerian velocity time scales $T_E$ from altimetry. For Lagrangian Chl time scales, $T_{L,\mathrm{Chl}}$ representative of a space bin is calculated from all enclosed float or drifter segments by integrating Eq. (7) and then space averaging (floats) or by integrating Eq. (8) (drifters). The latter is also done for Eulerian time scales $T_{E,\mathrm{Chl}}$ from GlobColour. Velocity standard deviations are computed by evaluating the segment standard deviation of each component, averaging them over all segments in a space bin to yield $\overline{\sigma_u}$ and $\overline{\sigma_v}$, and then taking $u' = 0.5\left(\overline{\sigma_u} + \overline{\sigma_v}\right)$. Finally, Lagrangian length scales are computed by multiplying together Lagrangian time and velocity scales as $L_L = u'T_L$ or $L_{L,\,\mathrm{Chl}} = u'T_{L,\,\mathrm{Chl}}$.

### 3.5 Integral length scales

There are 20 x 20 altimetric geostrophic velocity anomalies per 5º x 5º space bin, per map. For each space bin we subtract from all points their scalar spatial mean (yielding $y'(\mathbf{r})$) and compute the distances between all possible pairs of points to construct a radial (isotropic) ACF for space lag $d$ at time $t$,

$$R(d) = C(d)/J(d) \tag{9a}$$





where

$$C(d) = \frac{1}{N(d)} \sum_{k=1}^{N(d)} \left[ y'(r_k) y'(r_k + d) \right] \tag{9b}$$

and

$$J(d) = \frac{1}{N(d)} \sqrt{\left[ \sum_{k=1}^{N(d)} y'(r_k)^2 \right] \left[ \sum_{k=1}^{N(d)} y'(r_k + d)^2 \right]}. \tag{9c}$$

$R(d)$ is integrated in the same manner as $R(\tau)$ to obtain an integral length scale $L$. This is repeated for one map per month over the study period (taken as the 15th of each month) and picks from each map are averaged to obtain $\overline{L_{E,u}}$ and $\overline{L_{E,v}}$. As with time scales, scales for velocity are defined as the average of zonal and meridional length scales, $L_E = 0.5 \left( \overline{L_{E,u}} + \overline{L_{E,v}} \right)$.

For computational reasons, our estimates of $L_{E,\mathrm{Chl}}$ in each space bin come directly from the variograms calculated from daily, mesoscale-isolating MODIS fields by Glover et al. (2018). While their analysis used variograms to interpret scale, it is easy to show (Appendix C) that a measure mathematically identical to integrating the ACF to its first zero crossing can be derived from the variogram parameters. To match the length scales defined by Middleton (1985), all Eulerian integral length scales (velocity and Chl) are multiplied by 2.

**3.6 Chl frequency spectra**

Due to limitations of the data, ACF-derived scales might be biased large (float scales due to large lag bins necessary for uneven sampling; Table 1) or short (ocean colour scales due to poor intra-segment estimates of the mean for sparse segments). Because the power spectrum $P(f)$ is the Fourier Transform of the ACF, a time series with an exponential ACF has power spectrum

$$P(f) = \frac{1}{\pi} \frac{T^{-1}}{T^{-2} + (2\pi f)^2}, \tag{10}$$

which depends only on decorrelation time $T$. This spectrum is characterized by a $f^0$ power law at low frequencies and $f^{-2}$ power law at high frequencies, with $(2\pi T)^{-1}$ setting the transition frequency. For an independent measure of the power spectrum (that does not rely on our estimated ACFs), spectra will be calculated on all Chl time segments (same segments used for ACFs) using the Lomb-Scargle method (Glover et al., 2011). The frequencies evaluated are the equivalent of the Fourier harmonics for the segment length and spectral estimates are retained for $f \leq 1/2 \langle \Delta t \rangle$, which is an effective Nyquist frequency based on the average separation between data points over the segment. For each platform, valid individual spectra are averaged together (ocean colour segments with at least 50 % good samples; float segments with at least 24 profiles).



### 4 Results and Discussions

### 4.1 Velocity analysis

### 4.1.1 Quasi-Planktonic Index, QPI

The QPI methodology is illustrated in Fig. 2, where we display two examples when a float was located near straining maxima at the intersection of attractive and repulsive flow features. These are regions of rapid tracer stretching as evidenced by the elongation of synthetic particle clouds and represent a good challenge for a profiling float to keep up with surface flows. In the case depicted in Fig. 2a where the time spanned by the three adjacent profiles is only 2 days, the QPI is small at 4.49 km. If we take 5 km as a threshold (as a compromise between ensuring a small QPI and having a usable amount of

data), the floats are generally not Lagrangian with respect to the surface flow, with QPI generally much larger than 5 km (Figure 3a). The net velocities experienced by the floats (by centered differencing their positions) are well-correlated to the surface geostrophic velocities projected onto the float track (zonal and meridional correlation coefficients of 0.79 over all floats) but are systematically smaller by a factor of 2.3 (2.4) for $u$ ($v$). The samples with QPI < 5 km tend to fall closer to a one-to-one line ('x' symbols in Fig. 3b). The good correlation suggests that the deeper mesoscale flows that the floats feel

are equivalent barotropic and the nature of their deviations from a surface Lagrangian trajectory are primarily in magnitude of displacement, not in direction. For comparison, net drifter velocities have a slope of nearly 1 with respect to surface geostrophic currents (Figure 3c) but are no better correlated to them (zonal correlation coefficient 0.75; meridional correlation coefficient 0.72). Scatter about the one-to-one line is partly due to ageostrophic effects.

The two examples in Fig. 2 differ markedly in their QPI, with a derivative time window increase from two to four days

corresponding to a QPI that is ~6 times larger and a float trajectory that is qualitatively different from surface trajectories. The profiling interval exhibits strong control on QPI, with QPI increasing nonlinearly with derivative time window (Figure 4). This relationship represents a combination of greater time for the float to experience more sluggish velocities at parking depth (seen in Fig. 3b), greater time afforded to integrate vertical shear, and the manifestation of two-particle dispersion statistics under quasi-geostrophic turbulence due to mismatch between float and synthetic particle initial locations.






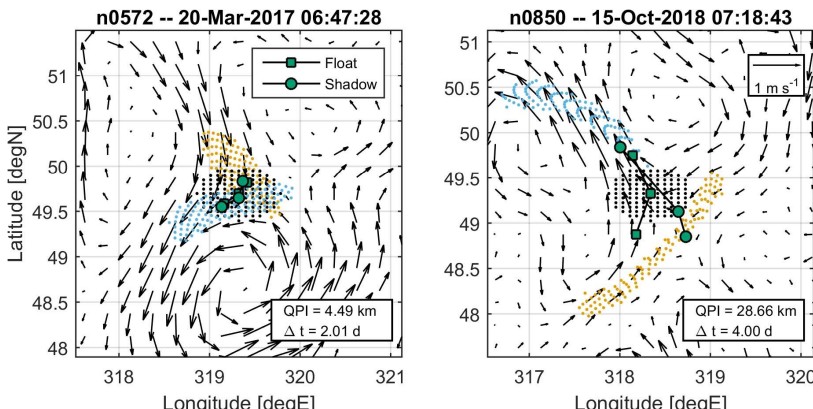

**Figure 2:** Example true (squares) and shadow (circles) float trajectories for **(a)** small Quasi-Planktonic Index (QPI) and **(b)** large QPI. In both panels, altimetric geostrophic currents are shown as vectors, initial particle locations are black dots, and final forward (backward) particle locations are blue (orange) dots. QPI and derivative time window indicated in lower right corner.

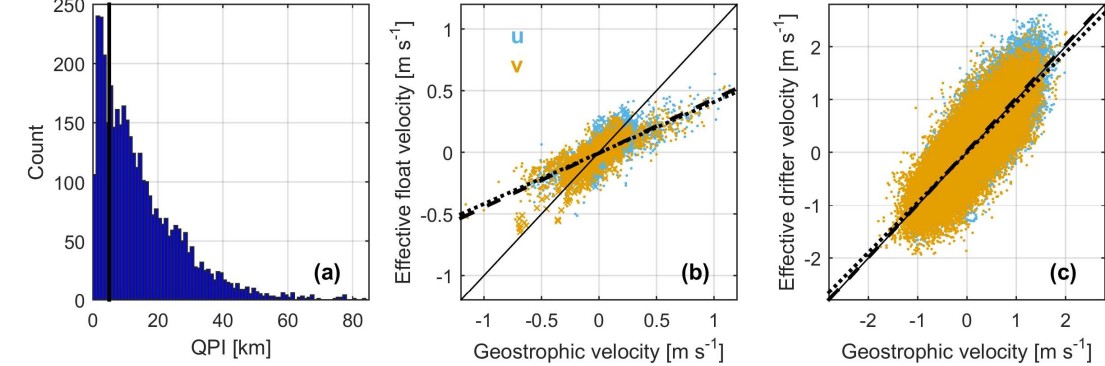

**Figure 3:** Overview of float velocities. **(a)** Histogram of Quasi-Planktonic Index (QPI) for all float profiles. Black line indicates 5 km. **(b)** Scatter plot of effective float velocity (centered difference position) against the total surface geostrophic velocity projected onto the float track. Blue (orange) dots are for zonal (meridional) velocity and similarly coloured x-symbols are for profiles with QPI < 5 km. Solid line is one-to-one line and dashed (dotted) line is least-squares regression line for zonal (meridional) velocity. **(c)** As in (b), but for drifters.

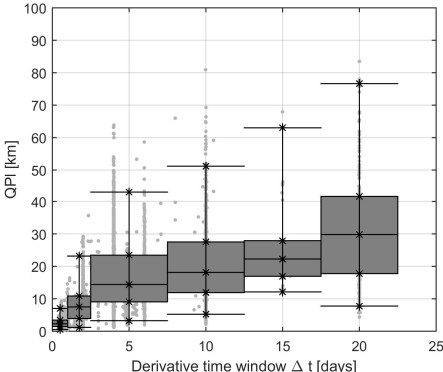


**Figure 4:** Scatter plot of Quasi-Planktonic Index (QPI) as a function of derivative time window (time spanned by three profiles) for all float profiles (dots) along with box and whisker plots indicating 2.5, 25, 50, 75, and 97.5 percentiles. Bins are [0, 1) days, [1, 2.5) days, and then span 5 days after that.

### 4.1.2 Integral time scales

Evaluating $T_L$ as a function of $u'$ (Figure 5a), drifters (open circles) and floats (solid circles) largely cluster into two separate regions, with drifters exhibiting greater velocity variance and a shorter time scale (~3 days compared to the ~5 days of floats). However, for space bins with multiple float segments, when we weight the individual scales by QPI$^{-2}$ (where the QPI is the segment-median; crosses connected by grey lines), we see the cluster of float points moves towards the cluster of drifter points (shorter $T_L$ and larger $u'$). In particular, the two rapid time sampling metbio* float segments (triangles in Fig. 5)

reside even closer to the cluster of drifter points than do the QPI$^{-2}$-weighted values from their host 5º x 5º bin. Nevertheless, they do not exactly reach the drifter scales of the host bin (open blue and orange circles). Lagrangian integral length scales (Figure 5c) generally exhibit a similar clustering of points with floats having shorter length scales and with the QPI$^{-2}$-weighted points moving towards the cluster of drifter points.

A similar relationship is found when examining $T_L/T_E$ as a function of $u'/c^*$ (Figure 5b). There are two clusters of points

(floats and drifters) which each reside on the theoretical curve of Middleton (1985), and the QPI$^{-2}$-weighted float values move along the curve towards the drifter values. The same is generally true for $L_L/L_E$ as a function of $u'/c^*$ (Figure 5d). These results suggest that drifters are generally in the "frozen-field" regime of turbulence while floats are primarily in the "fixed-float" regime, effectively acting as moorings. We found the floats to exhibit a continuum of behaviour, with segments characterized by a smaller median QPI (trajectories more similar to a surface Lagrangian trajectory) residing closer to

drifters in $T_L/T_E$ and $L_L/L_E$ versus $u'/c^*$ space. Overall, this is consistent with the currents experienced by the floats being just as well-correlated to surface geostrophic currents as are the currents experienced by surface drifters (Figure 3), suggesting that $T_E$ calculated from mesoscale geostrophic velocity anomalies derived from altimetry is a good estimate for both the surface and deep flow and the mesoscale currents are generally equivalent barotropic with only a small horizontal





wavenumber attenuation over depth. Floats in general traverse the eddy field more sluggishly than a surface parcel (but in a

manner still aligned with the surface flow) and this will impact how they sample a reactive tracer.

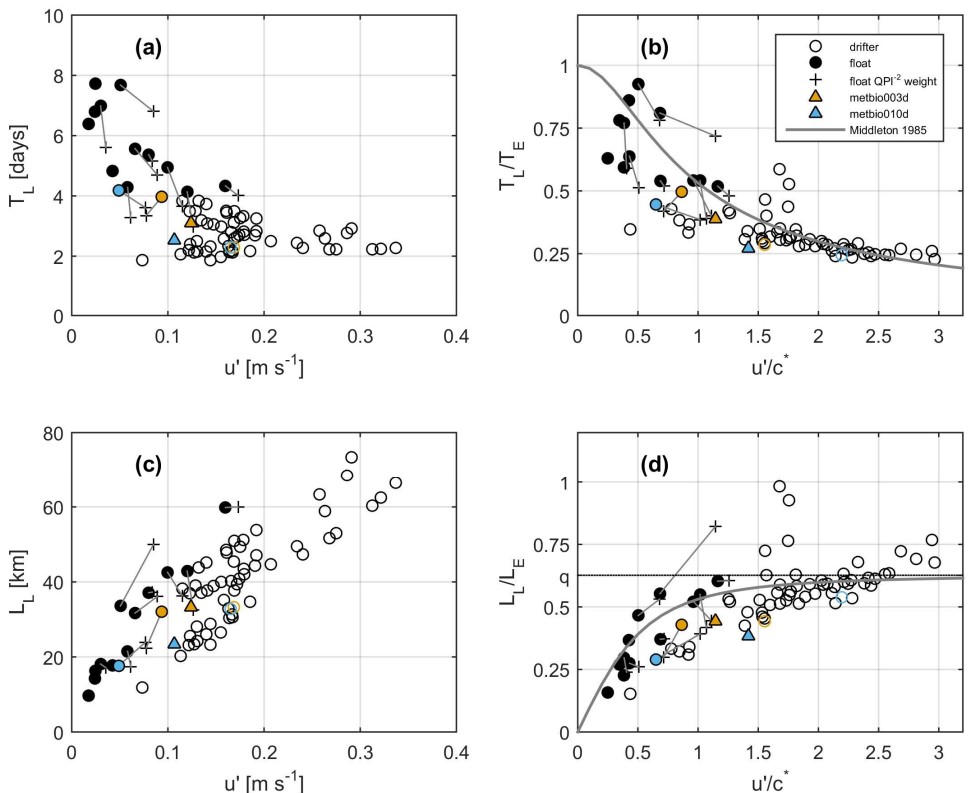

**Figure 5:** Scatter plots of velocity time and length scales for Lagrangian ($T_L$ and $L_L$) and Eulerian ($T_E$ and $L_E$) frames. **(a)** $T_L$ as a function of scale for the mesoscale eddy velocities, $u'$; **(b)** $T_L/T_E$ as a function of $u'/c^*$, where $c^*$ is the evolution speed for the eddy field, $c^* = L_E/T_E$; **(c)** $L_L$ as a function of $u'$; **(d)** $L_L/L_E$ as a function of $u'/c^*$. Hollow (filled) circles come from surface drifters (Bio-

Argo floats) and crosses weight the Bio-Argo float-derived scales by QPI⁻², with light grey lines connecting the weighted and unweighted values. Coloured triangles come from two time segments from two floats (metbio003d – orange; metbio010d – blue) with frequent and shallow profiling. The float and drifter circles coloured in the same manner are scales corresponding to the 5° x 5° bins hosting those two segments. Solid line indicates the theoretical relation of Middleton (1985).

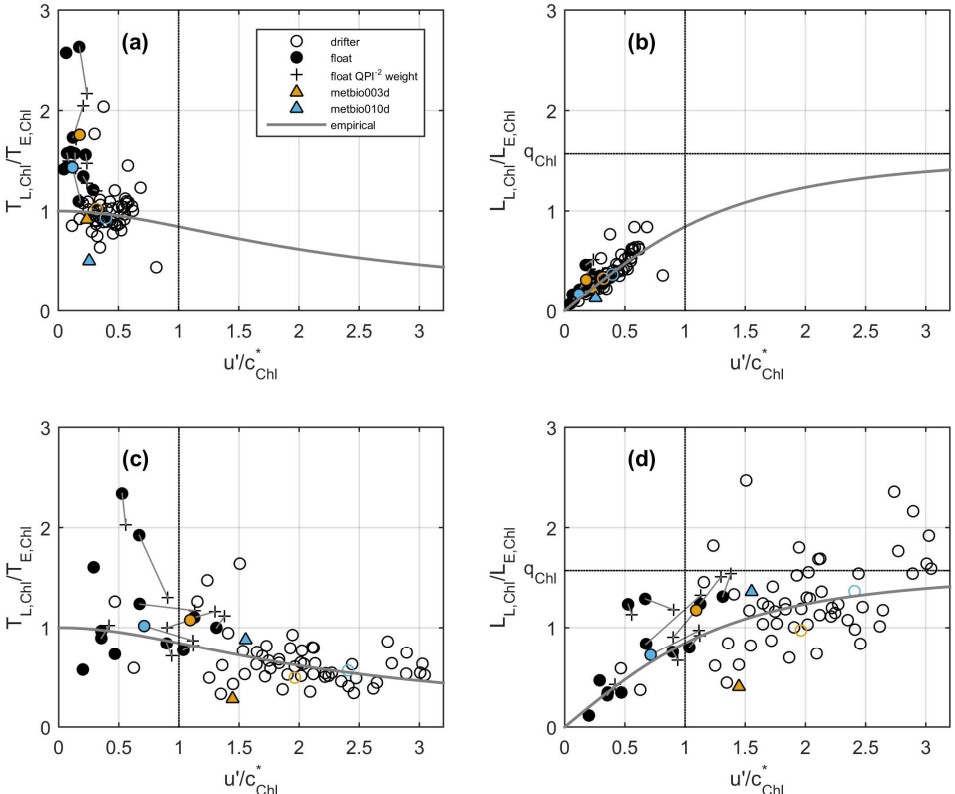

**Figure 6:** Scatter plots of chlorophyll time and length scales for Lagrangian ($T_{L,\mathrm{Chl}}$ and $L_{L,\mathrm{Chl}}$) and Eulerian ($T_{E,\mathrm{Chl}}$ and $L_{E,\mathrm{Chl}}$) frames. **(a)** $T_{L,\mathrm{Chl}}/T_{E,\mathrm{Chl}}$ as a function of $u'/c^*_{\mathrm{Chl}}$ from anomalies relative to smoothed subtrahend, where $u'$ is scale for the mesoscale eddy velocities and $c^*_{\mathrm{Chl}}$ is the evolution speed for the chlorophyll field, $c^*_{\mathrm{Chl}} = L_{E,\mathrm{Chl}}/T_{E,\mathrm{Chl}}$ ; **(b)** $L_{L,\mathrm{Chl}}/L_{E,\mathrm{Chl}}$ as a function of $u'/c^*_{\mathrm{Chl}}$ from anomalies relative to smoothed subtrahend; **(c)** and **(d)** are as in (a) and (b) but for anomalies relative to climatology subtrahend. Symbols are identified in legend and are exactly as in Fig. 5. Grey curves are from Eq. (11) with $q_{\mathrm{Chl}} = \pi/2$ and $s = 2$.

## 4.2 Chlorophyll analysis

### 4.2.1 Chlorophyll scales relative to smoothed subtrahend

Relative to the smoothed subtrahend, both floats and drifters experience $u'/c^*_{\mathrm{Chl}} < 1$ even though drifters move faster than floats, so the Chl field is evolving faster than either platform moves (Figure 6). This suggests that the primary balance in the material derivative should be an approximate correspondence of the Eulerian and Lagrangian tendency terms, EUL ≈ LAG



(Eq. (5)). We see that this is the case, for example, for drifters with ocean colour in the two space bins that host the rapidly
sampling metbio* segments (Figure 7a-b; see locations as bolded bins in Fig. 1). The eddy advection term (ADV) is about 25
% the magnitude of either EUL or LAG and is presumably less important. Drifters sample EUL ≈ LAG while floats sample
LAG < EUL, except for the rapidly profiling metbio* segments during which the float is behaving most like a drifter and
EUL ≈ LAG. Since the primary balance is EUL ≈ LAG, $T_{L,\text{Chl}}/T_{E,\text{Chl}}$ for drifters is approximately 1 in every space bin

(Figure 6a). The ratio is likely also ~1 for floats, but our ratios are biased large because coarse float time sampling demands
using larger lag bins in the ACF (Table 1), causing structure of the ACF at small lag to be poorly resolved. We know that the
float scales are biased large because $T_{L,\text{Chl}}/T_{E,\text{Chl}}$ must approach 1 as $u' \to 0$ (the observer is stationary), but that is not the
case. Averaged float scales weighted by QPI⁻² are shorter, approaching drifter with ocean colour scales. For both platforms,
because EUL ≈ LAG, $L_{L,\text{Chl}}/L_{E,\text{Chl}} < 1$ everywhere (Figure 6b). That is because neither platform has enough time to traverse

$L_{E,\text{Chl}}$ before Chl becomes decorrelated. This is corroborated by counting the number of ocean colour pixels traversed by a
drifter over $T_{L,\text{Chl}}$, which is ~1. The composite frequency spectra for both platforms (Figures 8a-b) reveal good agreement
with the model of an exponential ACF with an *e*-folding time $T_{L,\text{Chl}}$ between 0.5 and 1 days (dashed reference curves). The
similar decorrelation time for both platforms is consistent with the interpretation that EUL ≈ LAG relative to this subtrahend
(the different motions of the two platforms do not matter) and is consistent with our interpretation that the float $T_{L,\text{Chl}}$

computed from the ACFs are likely biased large and in reality are closer to the $T_{L,\text{Chl}} \sim 1$ day of drifters with ocean colour.

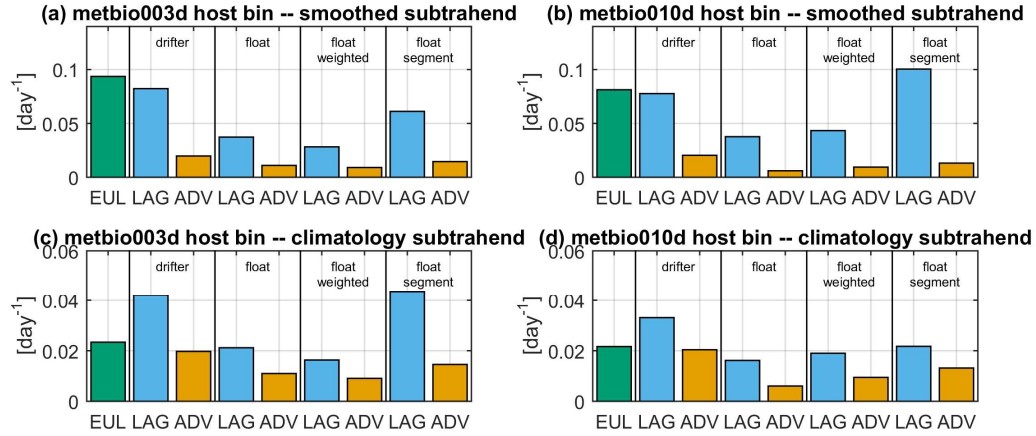

**Figure 7:** Magnitude of the material derivative terms for log-transformed surface chlorophyll using Eq. (5) in the two space bins hosting
the rapidly profiling metbio* segments. The Eulerian tendency (EUL, bluish green) term is plotted on the left of each panel, and the
Lagrangian tendency (LAG, blue) and eddy advection (ADV, orange) terms are displayed separately for drifters, floats, floats with average
scales weighted by QPI⁻², and the metbio* segments themselves. Panels **(a)** and **(b)** are calculated from Chl anomalies relative to the





smoothed subtrahend and panels **(c)** and **(d)** are calculated from Chl anomalies relative to the climatological subtrahend. Panels **(a)** and **(c)** are for bin hosting the metbio003d segment and panels **(b)** and **(d)** are for bin hosting the metbio010d segment.

### 4.2.2 Chlorophyll scales relative to climatological subtrahend

Relative to the climatological subtrahend, floats experience $u'/c_{\mathrm{Chl}}^{*} \leq 1$ while drifters experience $u'/c_{\mathrm{Chl}}^{*} > 1$. So, drifters

sample Chl space-time fields by traversing mesoscale Chl structures of diameter $L_{E,\,\mathrm{Chl}}$ while floats sample Chl somewhat more like a fixed observer (Figure 6c-d). Drifters measure $T_{L,\mathrm{Chl}}/T_{E,\mathrm{Chl}} < 1$ while floats measure $T_{L,\mathrm{Chl}}/T_{E,\mathrm{Chl}} \approx 1$. Note that this distinction between platforms is similar to what we saw regarding how they each sample the velocity field (compare Figures 5b,d and 6c-d), the meaning of which will be discussed in more detail later. With $u' > c_{\mathrm{Chl}}^{*}$ for drifters (and $u'$ occasionally close to $c_{\mathrm{Chl}}^{*}$ for floats), we expect effects of advection to be important. We see this to be the case, for example,

in the two space bins hosting the rapidly profiling metbio* segments (Figures 7c-d). For drifters with ocean colour, EUL ≈ ADV and each is about 50-60 % of LAG in those space bins, confirming that all terms are important. For floats, the ADV term is smaller than it is for drifters (because floats have smaller velocity variance) except for the rapidly profiling metbio* segments which have ADV close in magnitude to drifter ADV. The LAG term for floats tends to be smaller than for drifters, except for the rapidly profiling metbio* segments, for which the term is large like it is for drifters.

Since EUL ≈ ADV for drifters, $T_{L,\mathrm{Chl}}/T_{E,\mathrm{Chl}} < 1$. This is because a portion of the Lagrangian Chl signal is decorrelated by the platform traversing lateral Chl gradients. For floats, because ADV is less important and LAG is reduced, $T_{L,\mathrm{Chl}}/T_{E,\mathrm{Chl}} \approx 1$, more closely approximating an Eulerian observer. Consistent with the importance of advection, drifters with ocean colour tend to experience $L_{L,\mathrm{Chl}}/L_{E,\mathrm{Chl}} \approx 1$ or even greater than 1 in some space bins (Figure 6d). This suggests that advection is driving the Lagrangian decorrelation since decorrelation is achieved after the drifter has travelled a distance greater than

$L_{E,\,\mathrm{Chl}}$. Drifters can experience $L_{L,\mathrm{Chl}} > L_{E,\mathrm{Chl}}$ because Lagrangian trajectories are often aligned with isolines of tracer (Lehahn et al., 2007), hence drifters do not move directly down-gradient. Floats, on the other hand, tend to sit on a linear one-to-one line in $L_{L,\mathrm{Chl}}/L_{E,\mathrm{Chl}}$ versus $u'/c_{\mathrm{Chl}}^{*}$ space (Figure 6d). This is because EUL, not ADV, plays a primary role in the decorrelation since floats move slower than the Chl field evolves, except for the fastest moving floats which plateau where the drifter points do in Fig. 6d. Note that averaged scales weighted by QPI$^{-2}$ move towards drifter scales in both $T_{L,\mathrm{Chl}}/T_{E,\mathrm{Chl}}$

and $L_{L,\mathrm{Chl}}/L_{E,\mathrm{Chl}}$ versus $u'/c_{\mathrm{Chl}}^{*}$ space (Figure 6c-d), with increased velocity variance and generally (but not exclusively) shorter $T_{L,\,\mathrm{Chl}}$ and longer $L_{L,\,\mathrm{Chl}}$. Note also that float-based results are not sensitive to ACF methodology (use of Eq. (7) or Eq. (8); Appendix D).

The frequency spectra (Figure 8c-d) deviate from the theoretical power spectra corresponding to an exponential ACF for any value of $T_{L,\,\mathrm{Chl}}$ (curves with $T_{L,\,\mathrm{Chl}} =$ 5 and 10 days are shown, approximately bracketing the calculated drifter and float scales





relative to this subtrahend) and instead take on an approximate $f^{-1}$ slope over calculated frequencies. Given the role of
ADV in driving decorrelation relative to this subtrahend, one possibility is that the Lagrangian $f^{-1}$ slope represents a
manifestation of an Eulerian wavenumber $k^{-1}$ slope associated with a passive tracer under QG turbulence (Smith and
Ferrari, 2009) as the platform traverses the approximately frozen field, but this cannot be confirmed. Regardless, an equal
spectral slope for both platforms is not inconsistent with our ACF-derived time scales but does mean that we cannot use the
spectra to corroborate them.

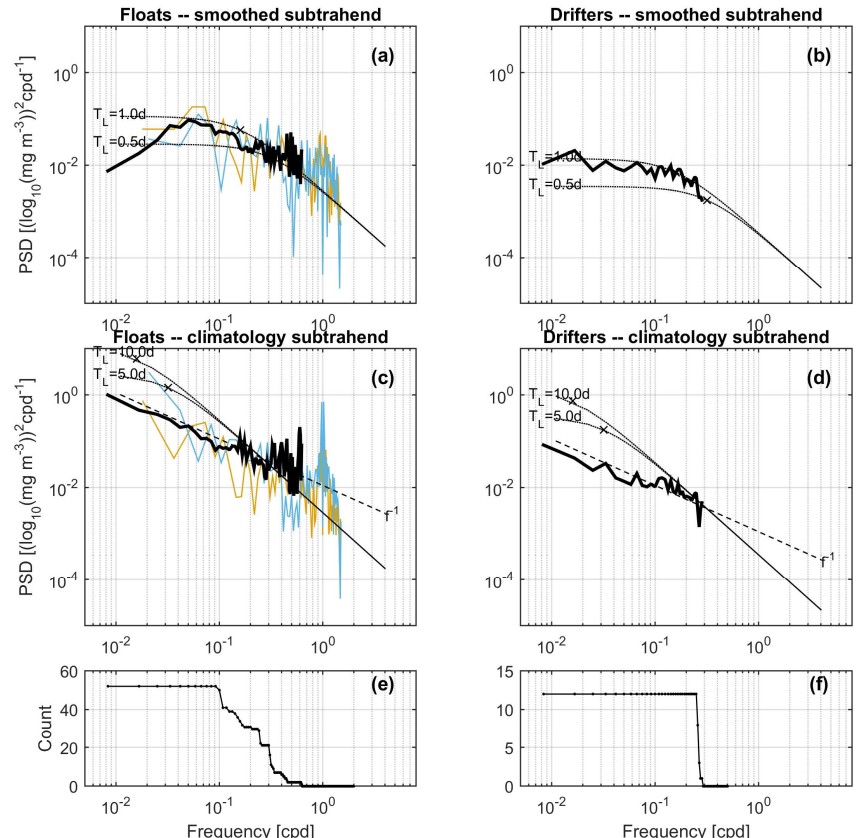

**Figure 8:** Averaged Lomb spectra for chlorophyll from all valid float (left) and drifter (right) segments as defined in Sect. 3.6. Spectra in top (middle) row are calculated from Chl anomalies relative to the smoothed (climatological) subtrahend. Bottom row gives counts of valid estimates per frequency. For float spectra, bold black line is the space-averaged spectrum while orange (blue) spectrum is from the
individual metbio003d (metbio010d) segment. Dotted reference spectra are theoretical spectra for an exponential autocorrelation function (ACF) with decorrelation time $T_L$ as labelled. Dashed spectra in middle panels give a -1 power law slope.





### 4.3 Biophysical interpretation of time and length scales

To interpret the meaning of these Chl scales, we compare them to the velocity scales. If stirring by the mesoscale velocity field is a primary driver of Chl variability, we expect the two variables to have similar Eulerian scales. Beginning with the

smoothed subtrahend, although the Eulerian space scales of Chl and velocity are similar (Figure 9b), the Eulerian time scales are unrelated (Figure 9a; squares), with $T_{E,\text{Chl}}$ fixed at ~2 days but $T_E$ spanning ~5-12 days. Along a trajectory, Chl decorrelates faster than velocity (Figure 9c) and over a shorter distance (Figure 9d) for both platforms, with floats experiencing longer decorrelation times of each variable but covering a shorter distance before becoming decorrelated. Eulerian velocities are calculated from satellite altimetry using a geostrophic balance, therefore containing mesoscale and

larger balanced flows, and we note that Lagrangian velocities are from drifter trajectories that were filtered in time to remove fluctuations shorter than 1.5 inertial periods, a procedure that presumably removes primarily unbalanced motions. Hence all velocity signals are likely dominated by balanced, mesoscale flows. The short time window of the Chl subtrahend (31 days) means that only rapid fluctuations (relative to inverse growth rates of mesoscale baroclinic instability or typical inverse phytoplankton accumulation rates) are retained. While tight coupling between division and loss rates tends to keep

accumulation rates $r$ low, abrupt changes in division rates due to rapid changes in environmental conditions can cause large-amplitude fluctuations in $r$ (Behrenfeld and Boss, 2018). Field studies in the subpolar North Atlantic observed near-surface accumulation rates of 0.47 day⁻¹ up to 0.77 day⁻¹ (Graff and Behrenfeld, 2018), corresponding to time scales of 1.3-2.1 days. Dynamically, submesoscale processes have length scales on the order of the deformation radius of the mixed layer ($O$(1 km)) and time scales on the order of an inverse inertial period ($O$(1 days)) (Mahadevan, 2016), scales smaller than an ocean colour

pixel and shorter than those of mesoscale dynamics. Hence, the EUL ≈ LAG balance for Chl relative to this subtrahend can be taken as dominance of ocean-colour pixel-scale variability in the retained signal due to either submesoscale processes or to biological sources/sinks.

For the climatological subtrahend, the Eulerian length scales are the same as relative to the other subtrahend, so again $L_{E,\text{Chl}} \approx L_E$. However, for this subtrahend, $T_{E,\text{Chl}} \approx T_E$ (Figure 9a; circles). This is consistent with mesoscale dynamics

setting Eulerian statistics of both velocity and Chl. Along a trajectory, Chl decorrelates more slowly than does velocity and Chl is correlated over a longer distance than is velocity. The Lagrangian time scale for floats is greater than it is for drifters because the reduced importance of the ADV term by less motion across mesoscale gradients (float $u' \leq c^*_{\text{Chl}}$) means Chl properties are retained longer. Equivalence of Eulerian length and time scales of velocity and Chl suggests that Chl scales are also dominated by balanced mesoscale dynamics, via advection. Because lateral motion and the ADV term are important, the

differing motions of floats and drifters means that they each sample different Chl signals relative to this subtrahend, with floats behaving somewhere between an Eulerian and a surface-Lagrangian observer. The exception is the two metbio* segments which sample Chl fields much like a surface-Lagrangian observer would.


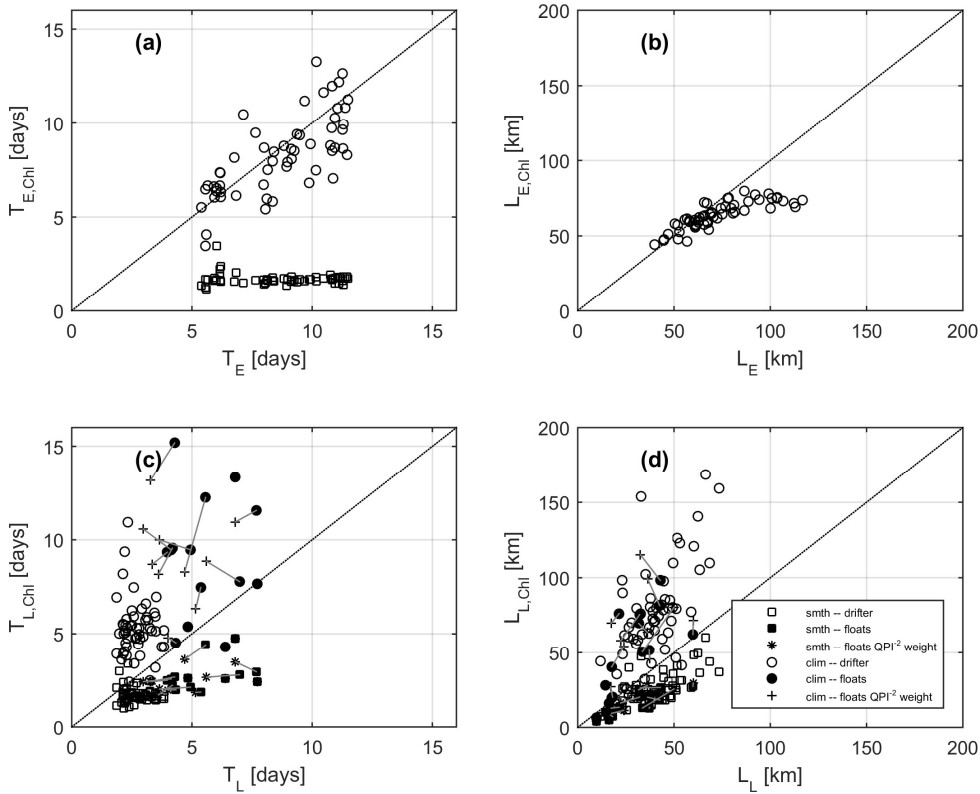

**Figure 9:** Time and length scales of chlorophyll (Chl) and velocity. **(a)** Eulerian time scales ($T_{E,\text{Chl}}$ versus $T_E$); **(b)** Eulerian length scales ($L_{E,\text{Chl}}$ versus $L_E$); **(c)** Lagrangian time scales ($T_{L,\text{Chl}}$ versus $T_L$); **(d)** Lagrangian length scales ($L_{L,\text{Chl}}$ versus $L_L$). Open circles are for drifters, filled circles are for floats, and crosses are for floats weighted by QPI$^{-2}$ where all scales are calculated from Chl anomalies relative to the climatology subtrahend. Open squares are for drifters, filled squares are for floats, and stars are for floats weighted by QPI$^{-2}$ where all scales are calculated from Chl anomalies relative to the smoothed subtrahend. Dotted lines are one-to-one.

### 4.4 Comparison with earlier estimates and a role for biology

Few studies have investigated Eulerian time scales of phytoplankton and even fewer have addressed Lagrangian time scales. Further, comparison of results across studies is complicated by myriad choices of data processing (e.g., subtrahends), intrinsic data resolution, and methodology (ACF or otherwise). In a series of studies, Denman and Abbott (1988, 1994) analyzed a scale-dependent decorrelation time by assessing the spatial coherence of ocean colour images separated in time and found Eulerian decorrelation times are generally less than about a week, being longer for 50-100 km wavelengths and shorter for 25-50 km wavelengths. Wavelengths smaller than 25 km are decorrelated after about a day. Comparing cross-





coherence of Chl and SST they conclude that physical stirring is the major driver of Chl variability at the mesoscale, a conclusion shared by Glover et al. (2018). In a more recent study, Kuhn et al. (2019) assessed Eulerian temporal decorrelation of numerically modelled biomass and found decorrelation times to generally be about 15 days, longer in regions of lower eddy-kinetic-energy and for larger phytoplankton size classes. Their longer decorrelation times may be

partially due to analyzing three-day averaged model fields. Our $T_{E,\,\mathrm{Chl}}$ fall in this broad range of values. The stark difference of values relative to the two subtrahends may be partially explained by the wavelength-based analyses of Denman and Abbott (1988), whereby the smoothed subtrahend is emphasizing pixel-scale (~25 km) variability, contributing to the short (~1-2 day) $T_{E,\,\mathrm{Chl}}$ compared to the climatology subtrahend (~5-12 day).

Estimates of $T_{L,\,\mathrm{Chl}}$ are rarer. Abbott and Letelier (1998) used bio-optical surface drifters in the California Current and found

$T_{L,\,\mathrm{Chl}}$ and $T_L$ in the open ocean are both about 2.5 days while $T_{L,\,\mathrm{SST}}$ is closer to 7 days, causing them to question the degree to which Chl behaves as a passive tracer. Boss et al. (2008) find much longer $T_{L,\,\mathrm{Chl}}$ at about 2 weeks using a profiling float. We also find $T_L \sim$ 2-3 days for drifters (longer for floats) but $T_{L,\,\mathrm{Chl}}$ is systematically larger or smaller depending on subtrahend (note Abbott and Letelier detrended float segments while Boss et al. only removed a scalar mean, the latter observing that seasonality dominated their longer decorrelation time). In a series of studies, Jönsson et al. (2009, 2011) used

synthetic particle trajectories and satellite ocean colour to quantify terms of the material derivative. They found the advective term is generally comparable in magnitude to the Lagrangian tendency and must be included. There are important differences in the magnitude and sign of each term, where a near-zero Eulerian tendency can be explained by a large Lagrangian tendency countering an advective term.

Analyses of Lagrangian series point to an importance of biological sources and sinks. Advection of phytoplankton across

spatial gradients of environmental conditions will affect dominant phenotypes (Lévy et al., 2014), with the relative time scales of physical parameters and physiological acclimation governing species succession (Zaiss et al., 2021). The LAG term is generally the largest term in the scaling of Eq. (5) (Figure 7), meaning sources and sinks of Chl are important. Relative to the smoothed subtrahend, as discussed earlier, the appropriate space scales are not resolved and an approximate balance of LAG ≈ EUL is achieved. But relative to the climatological subtrahend, the magnitudes of LAG and ADV tend to scale with

each other, being largest for drifters and the metbio* segments and smallest for the unweighted float segments (Figure 7). By moving slower than phytoplankton patches, floats underestimate biological sources (reducing LAG) and, that being so, there is less Chl variance apparently advected (reducing ADV, where we interpret platform speed as flow speed). Note that this is consistent with the interpretation posed in Sect. 4.3: biological sources and sinks are contained in the information content of ocean colour space-time fields and will project onto an Eulerian tendency or an advective term depending on the relative

rates at which a phytoplankton patch and observer move.

Finally, comparing Lagrangian and Eulerian timescales of Chl and of phytoplankton biomass (from backscattering) reveals regional discrepancies, confirming that Chl contains an acclimation component; however, each variable has a similar ratio of



Lagrangian-to-Eulerian timescales, confirming that they are sampled by a moving observer in the same manner and giving us confidence in our Chl-based biophysical interpretation (Appendix E).

### 4.5 Empirical relationship between chlorophyll Lagrangian and Eulerian scales

The plots of $T_L/T_E$ and $L_L/L_E$ as functions of $u'/c^*$ (Figure 5b,d) and $T_{L,\text{Chl}}/T_{E,\text{Chl}}$ and $L_{L,\text{Chl}}/L_{E,\text{Chl}}$ as functions of $u'/c^*_{\text{Chl}}$ (Figure 6c-d) have the same general shape. A similar functional dependence is not surprising given correspondence of Eulerian length and time scales for both velocity and Chl (Figure 9a-b), with only subtle differences in the Lagrangian length and time scales of each variable (Figure 9c-d) accounting for differences in the underlying functions. For this reason, we posit a mesoscale relationship for Chl may take the same form as that Middleton (1985) derived for velocity,

$$T_{L,\text{Chl}}/T_{E,\text{Chl}} = F\left(\alpha_{\text{Chl}}\right) = q_{\text{Chl}}\left(q^s_{\text{Chl}} + \alpha^s_{\text{Chl}}\right)^{-1/s} \tag{11a}$$

$$L_{L,\text{Chl}}/L_{E,\text{Chl}} = G\left(\alpha_{\text{Chl}}\right) = \alpha_{\text{Chl}}q_{\text{Chl}}\left(q^s_{\text{Chl}} + \alpha^s_{\text{Chl}}\right)^{-1/s}, \tag{11b}$$

where the constant $q_{\text{Chl}}$ sets the asymptotic value at large $\alpha_{\text{Chl}}$ and the exponent $s$ sets how rapidly $G$ transitions between linear and constant behaviour. In Middleton's (1985) velocity scaling (Eq. (3)), the asymptotic value $q$ is less than 1 because the variable being decorrelated is the same variable causing the decorrelation and $L_L$ never reaches $L_E$. For Chl, one possibility is to set $q_{\text{Chl}} = 1$ so that as ADV dominates the Lagrangian decorrelation $\left(u'/c^*_{\text{Chl}} \to \infty\right)$ we have $L_{L,\text{Chl}} \to L_{E,\text{Chl}}$. However, we have seen that $L_{L,\text{Chl}} > L_{E,\text{Chl}}$ routinely. A better choice (purely empirically) seems to be $q_{\text{Chl}} = \pi/2$ so that as ADV dominates the Lagrangian decorrelation we have $L_{L,\text{Chl}}/L_{E,\text{Chl}} \to \pi/2$ (grey curves in Fig. 6). For simplicity we set $s = 2$.

We assume that advection across Chl gradients is a primary factor in setting Lagrangian decorrelation (at least for the portion of the domain where $u'/c^*_{\text{Chl}} > 1$ which sets the asymptotic value), allowing us to appeal to an idealized geometry of mesoscale eddies. Following the convention of Middleton (1985), $L_E$ equals twice the decorrelation length and is effectively a mesoscale eddy diameter. From this perspective, $L_{E,\text{Chl}}/2$ serves as a radius of curvature for the maximal $L_{L,\text{Chl}} = \pi L_{E,\text{Chl}}/2$ (half a circumference) of a parcel traversing the perimeters of one or more mesoscale eddies. Half a circumference may be a meaningful decorrelation length in the scenario where a mean Chl gradient is stirred by an eddy of diameter $L_E$, partitioning the eddy into two hemispheres of high and low Chl, respectively (e.g., see Fig. 2a of Gaube et al. (2014)), or in the scenario that a float is traversing the perimeters of eddies characterized by monopolar cores of alternating uniform high and low Chl.



## 5 Conclusions

We analyzed the Lagrangian-Eulerian statistics of velocity and chlorophyll (Chl) as measured by Bio-Argo floats and as represented by satellite ocean colour (GlobColour) projected onto surface drifter tracks. Lagrangian statistics of velocity satisfy the Middleton (1985) relations, with drifters in a "frozen field" regime (spatial Eulerian decorrelation drives temporal Lagrangian decorrelation) and floats in a "fixed-float" regime (Eulerian tendency drives decorrelation). However, there is a continuum of behaviour with segments weighted by the inverse square of the Quasi-Planktonic Index (QPI) – a metric

quantifying similarity of float trajectory to a surface geostrophic trajectory – approaching the frozen-field limit. This is made possible by the mesoscale flows being approximately equivalent-barotropic with small horizontal wavenumber attenuation over depth. Given the space-time resolution of our ocean colour product (and typical float time sampling), both floats and drifters sample anomalies relative to the smoothed subtrahend as fixed observers, suggesting that at periods shorter than 31 days, Lagrangian Chl variability is dominated by ocean-colour-pixel-scale processes. Analysis of a finer ocean colour

product and a faster sampling observer is necessary to elucidate biophysical mechanisms (e.g., submesoscale sources and sinks). However, relative to a climatological subtrahend, Eulerian decorrelation time and length scales of Chl match those of velocity, suggesting mesoscale physical dynamics are important in setting plankton distributions as suggested by earlier studies (Denman and Abbott, 1994; Glover et al., 2018). The ratio of Lagrangian to Eulerian length scales for chlorophyll, $L_{L,\mathrm{Chl}}/L_{E,\mathrm{Chl}}$, depends on how fast the observing platform moves relative to how fast the Chl field evolves $\left(u'/c^*_{\mathrm{Chl}}\right)$,

following an empirical curve that appears to have the same functional form as that for velocity but with the asymptotic value replaced by scalar $q_{\mathrm{Chl}} \geq 1$, a value consistent with stirring by mesoscale eddies. Like Lagrangian statistics of velocity, floats reside closer to a "fixed-float" regime relative to Chl space-time variability and, as with velocity, weighting floats by QPI[-2] moves their values in parameter space towards drifter values along the empirical curve. Qualitatively speaking, the slower horizontal motion of floats relative to surface Chl patch speed means that advection across horizontal Chl gradients is

reduced and intra-patch biological sources and sinks are underestimated, both leading to longer time scales $T_{L,\mathrm{Chl}}$ and smaller Lagrangian tendency (LAG) and eddy advection (ADV) terms compared to a surface Lagrangian observer.

Practically speaking, a key question is whether floats should be analyzed in a Lagrangian framework. To answer this question, one approach is to compare the QPI to length scales of interest. With frequent profiling, QPI can be much smaller than length scales $L_{L,\mathrm{Chl}}$ and $L_{E,\mathrm{Chl}}$ (compare Fig. 3a and 9b) and the profiling intervals associated with such small QPI

(Figure 4) tend to be smaller than, but of the same order as, $T_{L,\mathrm{Chl}}$. Floats tend to have $T_{L,\mathrm{Chl}}$ about twice the value for drifters (Figure 9c) and $u'/c^*_{\mathrm{Chl}}$ about half as large (Figure 6), so they generally sample more like an Eulerian observer. Weighting by QPI[-2] does move float scales in the right direction, but not substantially so. The most promising segments are the two metbio* segments, which move the fastest relative to $c^*_{\mathrm{Chl}}$ and have amongst the shortest $T_{L,\mathrm{Chl}}$ compared to other float estimates (Figure 6), as well as largest ADV and LAG terms (Figure 7). Regardless, the QPI provides direct





quantification of Lagrangian similarity over the time spanned by a centered-difference derivative (and can be evaluated for arbitrary profile sequences), so even if mismatch is integrated over time, individual estimates of phytoplankton accumulation rate $r(\mathbf{x}(t),t)$ can be good in moving snapshots.

More and more Bio-Argo floats are being deployed (e.g., GO-BGC array; Roemmich et al., 2019; https://biogeochemical-argo.org/), and their canonical profiling interval is 10 days (derivative time window $\Delta t = 20$ days in Fig. 4). Our results show

that such a long profiling interval typically corresponds to QPI values that are of the same order as $L_{E,\text{Chl}}$ (Figures 4 and 9b). Floats tend to exhibit $T_{L,\text{Chl}}$ near the 10 day threshold (Figure 9c), but actual surface $T_{L,\text{Chl}}$ for mesoscale anomalies is shorter at about 5 days (Figure 9c) since floats underestimate surface horizontal motion (so that $L_{L,\text{Chl}}$ from floats would be less than $L_{E,\text{Chl}}$; Figure 6d). The relationship between profiling interval and QPI (Figure 4) suggests that the ability of profiling floats to be Lagrangian with respect to the surface flow is, to some degree, controllable. To obtain near-surface

behaviour (e.g., as exhibited by the two metbio* segments), $\Delta t$ needs to be very small (multiple profiles per day), which is incompatible with the GO-BGC mission of maximizing time series length because of the power draw and risk of biofouling. Nevertheless, there may be value in interspersing periods of frequent, shallow profiling, for example when located in a biophysically relevant feature (straining front or mesoscale eddy). These brief windows of $r(\mathbf{x}(t),t)$ should be more accurate than those obtained by space-averaging float time series (destroying quasi-Lagrangian information) and could be

used to inform a process-based understanding of phytoplankton variability.

**Appendix A**

The Quasi-Planktonic Index (QPI) is a single number that quantifies the similarity of a float trajectory to the best-fit synthetic trajectory (computed by advection of synthetic particles with altimetric total geostrophic currents) over the temporal footprint of a centered difference derivative (three total float profiles). The procedure is repeated for each float

profile $i$. We first create a disk of particles about $\mathbf{r}_i = (\text{lon}_i, \text{lat}_i)$ by making a rectangle about $\mathbf{r}_i$ with particles spaced zonally by $\delta_0 / \cos(\text{lat}_i)$ out to $\text{lon}_i \pm \delta_{\text{total}} / \cos(\text{lat}_i)$ and meridionally by $\delta_0$ out to $\text{lat}_i \pm \delta_{\text{total}}$ and then retaining those values with spherical distance less than or equal to 0.3°. We chose $\delta_0 = 0.05°$ and $\delta_{\text{total}} = 0.3°$. The disk of $K$ particles is advected by the altimetric total geostrophic currents forward in time with a fourth-order Runge-Kutta scheme at an hourly time step from their initial position at $t_i$ to the first hour past float time $t_{i+1}$. Velocity fields at intermediate time steps are obtained through

linear interpolation and particle velocities are updated through bilinear interpolation in space. For the $k^{\text{th}}$ trajectory beginning at float time $t_i$ its position at float time $t_{i+1}$ is obtained by linearly interpolating its positions at the two surrounding hourly time steps. Similarly, the disk of $K$ particles is advected backward in time with the same fourth-order Runge-Kutta scheme





and hourly time step out to the first hour prior to $t_{i-1}$. For the $k^{\text{th}}$ trajectory beginning at float time $t_i$, its position at float time $t_{i-1}$ is obtained by linearly interpolating its positions at the two surrounding hourly time steps.

The advection gives $K$ sets of three positions at each float step $i$. The penalty function for trajectory $k$ at float step $i$ measures the average distance between the $k^{\text{th}}$ synthetic trajectory and the true float trajectory over the three time steps that constitute the centered difference:

$$S_{i,k} = \frac{1}{3} \sum_{j=-1}^{+1} \text{dist}\left(\mathbf{r}_{\text{float}}(i+j), \mathbf{r}_k(j)\right) \tag{A1}$$

where $\mathbf{r}$ is the position in latitude-longitude coordinates and $\text{dist}(\cdot)$ is a measure of distance on the sphere. The QPI for time

step $i$ is

$$\text{QPI}_i = \min_{k \in K} S_{i,k} \tag{A2}$$

and has units of kilometres. The corresponding shadow trajectory is the three-element trajectory $\mathbf{r}_k$ for the $k$ that minimizes $S$.

**Appendix B**

The first subtrahend ("smoothed") is meant to replicate that used by Glover et al. (2018) so that we can obtain Lagrangian and Eulerian time scales consistent with their Eulerian space scales. For Eulerian data, we perform a 3-D convolution of all GlobColour space-time fields with a 3-D filter defined as a 31-day Hamming window in time and a 2-D Gaussian in space with a 1º full-width-half-maximum and a 2º cutoff. Just as the total GlobColour data are bilinearly interpolated onto drifter tracks, the subtrahend is interpolated in the same manner and the two are differenced to obtain Lagrangian Chl anomalies.

Finally, for a float Chl subtrahend, we perform a weighted running average of each float Chl series with weights defined by a 31-day Hamming window. This approach to smoothing the float data is less than desirable because it constitutes a Lagrangian subtrahend whereas the drifter anomalies are about an Eulerian subtrahend projected onto a Lagrangian trajectory. Further, this approach cannot implement an equivalent spatial smoothing in the subtrahend for the float data since they come from a single Lagrangian trajectory (see discussion below). The time filter component of the "smoothed"

subtrahend is effectively a low-pass filter and, with a cutoff of 31 days, we found it to retain a sizable portion of the intraseasonal (and perhaps mesoscale) variance. This is particularly the case for the float data, where uneven spacing means that the effective cutoff frequency will vary depending on the sparsity of sampling in a 31-day window since data points near the center of the window are weighted more.

    As an alternative, the second subtrahend strictly isolates climatological variability ("climatology"). The spatial footprint of

the filter is the same, but the time filtering is performed about a day-of-year coordinate rather than an absolute date coordinate (for example, 1 January of every year is regarded as having the same time coordinate). Specifically, for Eulerian data, we perform a 4-D convolution of all GlobColour space-time fields with a 4-D filter defined as a 31-day-of-year





Hamming window in time, a 2-D Gaussian in space with a 1º full-width-half-maximum and a 2º cutoff, and a boxcar function (equal weights) across years. Again, the subtrahend is bilinearly interpolated onto the drifter tracks.

The major difference for the "climatological" subtrahend concerns how the floats are treated. There are not enough co-located floats to create a climatological subtrahend from float-measured Chl, so we construct a float subtrahend from the filtered GlobColour data (Eulerian subtrahend). Because floats measure Chl year-round while GlobColour contains significant seasonal gaps, we first regress annual and semiannual cosines to the Eulerian subtrahend in each pixel as an interpolant. While inclusion of higher harmonics would yield a better fit in some regions (especially in regions with complex

seasonal cycles such as those with spring and fall phytoplankton blooms), we stick to only the first two harmonics since gaps in the Eulerian subtrahend of up to 6 months would lead to significant overshoot if higher harmonics were included. The resulting field is then projected onto the float tracks (with a simple nearest neighbour approach in space and time). Finally, because the dynamic range of GlobColour Chl is different from that of the MLD-averaged float Chl series, the projected subtrahend needs to be scaled before it can be removed from the float data. So, as a final step we regress the projected

subtrahend against actual float Chl (all samples across all floats) to obtain a best estimate of seasonal cycle amplitude.

   A summary of the subtrahends is provided in Table B1, and a visual example of time series is shown in Fig. B1. Anomalies about the "climatology" subtrahend contain substantially more low-frequency variance, including intraseasonal variability but also interannual variability due to year-to-year variations in phasing and amplitude of blooming. The "climatology" subtrahend has the added benefit of being applied to the floats and drifters in an equivalent manner as an Eulerian subtrahend

projected onto a Lagrangian trajectory. It is fair to question whether the "smoothed" subtrahend as computed for floats is comparable to the "smoothed" subtrahend as computed for ocean colour pixels and drifters because the former does not include explicit spatial smoothing, only a combined space-time Lagrangian smoothing. As a test, Fig. B1 additionally includes the Eulerian "smoothed" subtrahend projected onto float tracks (via nearest-neighbour interpolant and regressed against float data to correct for dynamic range) for periods where we have overlapping data. In general, the projected

Eulerian "smoothed" subtrahend closely agrees with the float "smoothed" subtrahend. Exact correspondence between float and satellite Chl is not expected since floats sample a parcel of water on the order of the size of the platform and gridded ocean colour products integrate information in space and time (see discussion in Yang et al. (2020)).

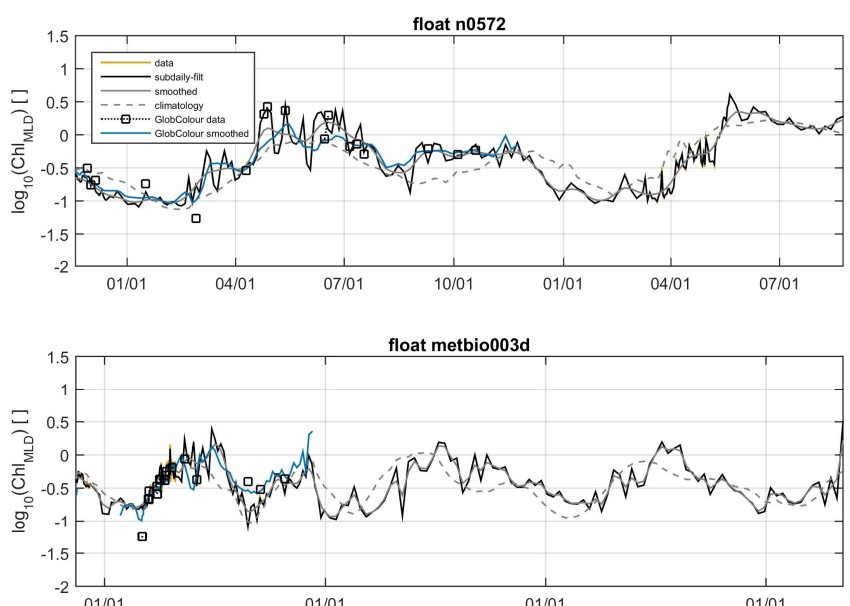

**Figure B1:** Example time series of log-transformed, MLD-averaged Chl from two floats (n0572 and metbio003d). Data are orange, data with sub-daily filter applied are black, the "smoothed" subtrahend is solid grey, and the "climatology" subtrahend is dashed grey. For evaluation, we also display the GlobColour data projected onto the float tracks (black squares) and the Eulerian "smoothed" subtrahend from GlobColour projected onto the float tracks (solid blue). Note that sample interval was generally over one day except for brief intervals so that sub-daily filtered and unfiltered series are generally identical (black curve generally over orange). Note also the variable
sample rate for metbio003d, illustrating how the effective cutoff period of the smoothed subtrahend can vary substantially from 31 days.

| Subtrahend name | Eulerian | Drifter | Float |
|---|---|---|---|
| Smoothed | 3-D convolution of all GlobColour space-time fields with 31-day Hamming window in time, 2-D Gaussian in space (1º FWHM, 2º cutoff) | Project Eulerian subtrahend onto drifter track | Along-track weighted running average with 31-day width Hamming window |
| Climatology | 4-D convolution of all GlobColour space-time fields with: 31-day Hamming window in day-of-year coordinate, 2-D Gaussian in space (1º FWHM, 2º cutoff), boxcar across years | Project Eulerian subtrahend onto drifter track | Regress annual + semiannual cosines onto Eulerian subtrahend in each pixel (to fill gaps), project resulting field onto float tracks, regress projected subtrahend against actual float Chl (all samples across all floats) to account for different dynamic ranges and potential biases between float and GlobColour Chl. |

**Table B1:** Summary of subtrahends.



**Appendix C**

Glover et al. (2018) calculate space scales of mesoscale Chl variability by calculating variograms of mesoscale Chl anomaly
fields in 5° x 5° space bins. They fit a spherical variogram model to their calculations by nonlinear regression, which in one
dimension is given by

$$\gamma(\delta) = \begin{cases} c_0 + (c_\infty - c_0)\left[\dfrac{3}{2}\dfrac{\delta}{a} - \dfrac{1}{2}\dfrac{\delta^3}{a^3}\right] & \text{for } \delta \leq a \\ c_\infty & \text{for } \delta > a \end{cases} \tag{C1}$$

where $c_0$ is the nugget (unresolved variance), $c_\infty$ is the sill (approximating the total variance), $a$ is the range (closely related
to the decorrelation length), and $\delta$ is the scale of space separation being evaluated. Intuitively, $a$ is closely related to the
integral decorrelation length (integral of spatial ACF to first zero crossing) but the two are not equal. The spatial
autocorrelation function $R(\delta)$ is related to the variogram by

$$R(\delta) = 1 - \frac{1}{C(0)}\gamma(\delta), \tag{C2}$$

where $C(\delta)$ is the autocovariance function. Using the fact that $C(0) = c_\infty$, the space lag $\delta_0$ at which the ACF $R = 0$ is
determined by rearranging Eq. (C2) to yield $c_\infty = \gamma(\delta_0)$, requiring $\delta_0 = a$. Thus, the integral of the ACF to the first zero
crossing is given by

$$L = \int_0^a R(\delta)d\delta = \int_0^a d\delta - \frac{1}{c_\infty}\int_0^a \gamma(\delta)d\delta \tag{C3}$$

which, after substituting in Eq. (C1), gives

$$L = \frac{3a}{8}\left(1 - \frac{c_0}{c_\infty}\right). \tag{C4}$$

Glover et al. (2018) report separate ranges $a_x$ and $a_y$ from MODIS data in the zonal and meridional directions, respectively.
Therefore, we calculate zonal and meridional integral length scales $L_x$ and $L_y$ from Eq. (C4) to define the Chl integral
length scale,

$$L_{E,\text{Chl}} = \left(L_x^2 + L_y^2\right)^{1/2}. \tag{C5}$$



**Appendix D**

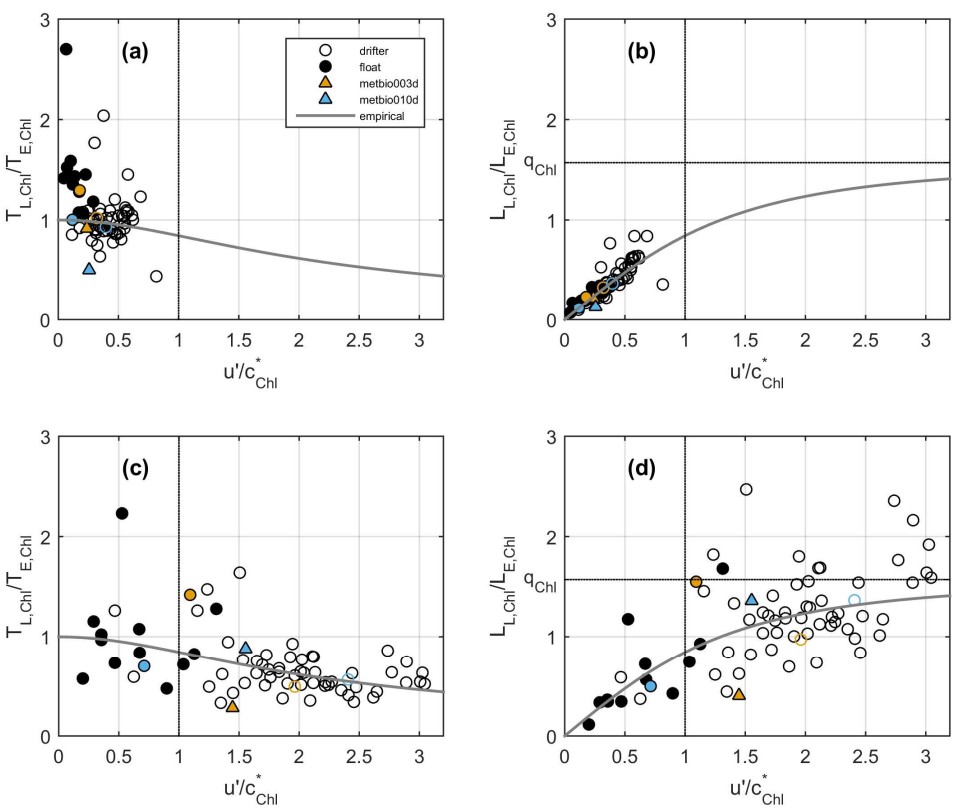

**Figure D1:** Exactly as in Fig. 6 but with float scales calculated by integrating Eq. (8) (space-composited ACF) instead of integrating Eq. (7) (individual segment ACF) and averaging over space.


**Appendix E**

In this paper we analyze chlorophyll since there is a history of geostatistical studies of that variable to build upon (e.g., Denman and Abbott, 1988, 1994; Doney et al., 2003; Glover et al., 2018; Eveleth et al., 2021). Chl is a complicated variable, containing a regionally (and seasonally) strong acclimation signal in addition to a biomass signal (Behrenfeld et al., 2005) that may imprint on our results. As a check, we evaluated Eulerian and Lagrangian timescales of phytoplankton carbon biomass ( $C_{phyto}$ ) and compared them to timescales of Chl. $C_{phyto}$ is derived from float- (at 700 nm) or satellite-measured (at 443 nm) backscattering (following Graff et al., 2015 Table 2 and assuming a spectral power law for backscattering of -0.78),





anomalies are computed relative to equivalently constructed subtrahends as for Chl (as described in Sect. 3.3), and its ACF is
integrated as for Chl (as described in Sect. 3.4). In general, Eulerian and Lagrangian timescales of Chl are longer than
timescales of $C_{\mathrm{phyto}}$ in the subtropics (as defined by the -0.10 m mean absolute dynamic topography contour; Della Penna
and Gaube, 2019), are shorter than timescales of $C_{\mathrm{phyto}}$ near coasts, and are approximately equal elsewhere, suggesting that
there is a regional acclimation signal built into Chl that affects its timescales in certain regions of the ocean. However,
importantly for this study, the relevant quantity $T_{L,\mathrm{Chl}}/T_{E,\mathrm{Chl}}$ is proportional to $T_{L,\mathrm{Cphyto}}/T_{E,\mathrm{Cphyto}}$ everywhere, meaning that
even if the biophysical drivers of variability in Chl and $C_{\mathrm{phyto}}$ vary regionally, the variables are sampled equivalently by a
Lagrangian (or quasi-Lagrangian) observer, giving us confidence in our Chl-based results.

**Data availability**

NAAMES float data are available from the University of Maine In-Situ Sound and Color Lab archive
(http://misclab.umeoce.maine.edu/floats/). *Sprof files for the non-NAAMES floats and *prof files for the NAAMES floats
(hydrographic variables only) are available from the IFREMER Argo Global Data Assembly Center (snapshot from
February 2021; http://doi.org/10.17882/42182#81474). Altimetry data (product
SEALEVEL_GLO_PHY_L4_REP_OBSERVATIONS_008_047, DT2018 reprocessing; https://doi.org/10.5194/os-15-1207-
2019) are archived by Copernicus Marine Environmental Services (https://marine.copernicus.eu/). The DT2018 version was
superseded by the DT2021 version at time of writing and is no longer accessible by url; however, data can be de-archived by
Copernicus upon request. GlobColour data (variables CHL1 and BBP using "merged" sensors, L3m, daily binning, 25 km
resolution, and the GSM algorithm) are from the R2019 processing available by web or ftp from ACRI-ST, France
(https://hermes.acri.fr). Drifter data are the quality-controlled six-hourly product available from the NOAA Global Drifter
Program (https://doi.org/10.25921/7ntx-z961), where we accessed the ASCII files on 14 September 2020
(https://www.aoml.noaa.gov/phod/gdp/interpolated/data/all.php).

**Author contribution**

DCM and SCD designed the study and DCM performed the analyses. ESB acquired and processed the NAAMES float data,
DMG provided the Chl variogram parameters, and ADP helped develop the QPI. DCM prepared the manuscript with
contributions from all co-authors.

**Competing interests**

The authors declare that they have no conflict of interest.





## Acknowledgements

This work was supported in part by the National Aeronautics and Space Administration (NASA) as part of the North Atlantic Aerosols and Marine Ecosystems Study (NAAMES; NASA grant 80NSSC18K0018). Additional support was provided to S. Doney and D. McKee from the University of Virginia. PG acknowledges the support of the NASA Ocean

Biology and Biogeochemistry Program and the PACE Science Team, NASA grant 80NSSC20M0202.

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
