# Peer review of "Lagrangian-Eulerian time and length scales of mesoscale ocean chlorophyll from Bio-Argo floats and satellites"

_Biogeosciences, 2022_

## Author Response (AR1)

**Lagrangian-Eulerian time and length scales of mesoscale ocean chlorophyll from Bio-Argo floats and satellites**

Darren C. McKee[1], Scott C. Doney[1], Alice Della Penna[2,3], Emmanuel S. Boss[4], Peter Gaube[5], Michael J. Behrenfeld[6], David M. Glover[7]

[1]Department of Environmental Sciences, University of Virginia, Charlottesville, VA, 22904, USA
[2]Institute of Marine Science, University of Auckland, Auckland, New Zealand
[3]School of Biological Sciences, University of Auckland, Auckland, New Zealand
[4]School of Marine Sciences, University of Maine, Orono, ME, USA
[5]Applied Physics Laboratory, University of Washington, Seattle, WA, USA
[6]Department of Botany and Plant Pathology, Oregon State University, Corvallis, OR, USA
[7]Department of Marine Chemistry and Geochemistry, Woods Hole Oceanographic Institution, Woods Hole, MA, USA

**Format of this document:**
Black = Referee Comments (posted online); Blue = Author Comments (posted online);
Green = Description of how manuscript has changed, following the Author Comments

**Author Comments in response to Referee #1**

The manuscript analyses decorrelation in time and space from both a Lagrangian and Eulerian perspective with the ultimate aim to estimate how well Argo float act as Lagrangian platforms. The motivation for the paper is sound and it addresses some very important questions. I'm excited to use the results from a published version of the MS in future studies and believe it to have a wide potential utility. There are, however, a coupleof major questions/concerns I need resolved before recommending publication.

Thank you for your close read and evaluation of our manuscript.

Before we address each of your comments individually, we would like to preface with an overview. The matters you bring up in your comments 1-3 are related by a single overarching point that we perhaps did not make clear enough in our original manuscript. Our intention in this study was to analyze time- and length-scales of mesoscale ocean chlorophyll variability (and velocity). This choice of scale dictated our choices of data products and filtering. Given this, our responses to your comments 1-3 are related.

1. The first equation suggests, to my understanding, that the Chl field is fixed in in space. This is a bold assumption that needs to be carefully motivated. I would have expected the advection decorrelation term to be applied to the Eulerian observer since Chl is advected with the velocity field. One could possibly argue that biomass might originate from stationary processes at for example seam mounds, but this is rather the exception than the rule. As a consequence, I expect that a Lagrangian sampling platform in general, with some specific exceptions, experiences longer temporal decorrelation time scales compared to the Eulerian observer. I'm willing to admit that I might have misunderstood Eq1 and the reasoning around it, but I don't think I'm the only one if so. This need to either be explained better or changed.

Equation (1) is simply a material derivative "budget" for chlorophyll and does not make any
assumptions about the properties of the chlorophyll field. In our reading, nothing about the
presentation of Equation (1) prescribes a behavior for the chlorophyll field. When the equation is
scaled (equation (5)), Eulerian scales are indeed used for the advection term. The discussion that
introduces Equation (1) (lines 84-107) is based on the velocity field, for which the theoretical
and observational studies cited show that, for velocity, it is true that $T_L < T_E$. However, prior to
conducting this study, it was not known whether there was any systematic relationship between
$T_{L,Chl}$ and $T_{E,Chl}$. By scaling Equation (1) with scales derived from the chlorophyll fields and
Lagrangian or Eulerian chlorophyll time series, we are able to consider how the movement of an
observer relative to "movement" of the chlorophyll fields ($u'$ versus $L_{E,Chl}/T_{E,Chl}$) influences the
Lagrangian decorrelation time.

Like you, we were surprised to find $T_{L,Chl} \leq T_{E,Chl}$. We suspect there are several possible origins
for this behavior. Firstly, this may be the manifestation of an observer moving across existing
gradients in the chlorophyll field, as would happen when a mesoscale eddy stirs a horizontal
gradient. The empirical curves in Figure 6 (from equation 11) would support this, as described in
Section 4.5. Another possibility is that chlorophyll may actually be conserved for longer along a
trajectory than our results would indicate: if patches are organized in small scale filaments that
are not fully resolved in an 0.25º product, the inability for a drifter-projected time series to
resolve such near-constant chlorophyll levels along a filament will result in an early temporal
decorrelation. The result that the ratio $T_{L,Chl}/T_{E,Chl}$ is approximately 1 relative to the smoothed
subtrahend (where sub-pixel variability probably dominates) while the ratio is less than 1 relative
to the climatology subtrahend (where larger and/or slower processes dominate) supports this
interpretation. We plan to add the preceding discussion to the manuscript.

Changes made: We added a section 4.6 which discusses how the relationship between $T_{l,Chl}$ and
$T_{e,Chl}$ may depend on the resolution of satellite data (basically, that these are results of mesoscale
variance). We did not change the presentation of Equation (1) after carefully reviewing section 2
and concluding that as currently written there is no presupposition of a relationship between $T_{l,}$
$_{Chl}$ and $T_{e,Chl}$ nor an assumption that the chlorophyll field is fixed.

2. The use of Chl fields with a 0.25° spatial resolution and the removal of sub- and mesoscale
variability weakens the study significantly. It is abundantly clear that submesoscale processes are
of first-order importance in controlling the variability of Chl, as mentioned in the MS and cited
publications by Amala Mahadevan or Marina Levy. A general analysis of decorrelation time-
and length scales can get away with using coarser grids by defining the domain of interest
carefully but this study doesn't have that luxury. One specific aim, as I understand, is to evaluate
the utility of float which requires the use of the highest resolution possible. I would have
preferred that a 1km product had been used (OC-CCI at 1km is for example available from
Plymouth Marine Laboratory), but I understand if a 4km product is used out of necessity. Aren't
the results quite dependent of the rather arbitrarily chosen 0.25° pixel size? How much would the
results differ if 0.125°, 0.5, or 1° pixels were used instead?

We acknowledge that submesoscale processes are of first-order importance in driving surface
chlorophyll variability. Further, we do believe that our results are dependent on the ocean color
pixel size. That being said, the choice of 0.25º was not arbitrary, and we believe our results are
still novel, meaningful, and useful. As indicated by the title (though perhaps the manuscript
needs to more clearly convey this), our interest is in studying the mesoscale chlorophyll field.

Our motivation for this interest is both a practical and intellectual matter.

As a practical matter, there is a tradeoff between resolving more variance and dealing with
increased gaps when moving to a finer resolution ocean color product. For the purpose of this
study, we chose to prioritize data coverage, leading us to select a blended, 0.25º product, and a
focus on the mesoscales. As an additional practical matter, given the relatively sluggish motion
of floats (as quantified in this study), they may not capture the full spectrum of submesoscale
processes. Given that we wanted to incorporate in-situ data in this study, we felt it best to focus
on mesoscale variations. The choice of product is also consistent with the grid size of the
Eulerian velocity field (0.25º altimetric geostrophic currents), and we aimed for consistency
since we compare the two variables.

Intellectually speaking, combined Lagrangian-Eulerian scales of chlorophyll are unknown at any
scale, and we believe that contributions at the mesoscale are useful. New results are still being
gleaned about geostatistics of the mesoscale chlorophyll field and their biophysical origin (e.g.,
Eveleth et al., 2021). We believe the results here stand on their own and our mesoscale study
may lay the groundwork for follow-up studies targeting the submesoscale, either utilizing a more
spatially or temporally expansive drifter and ocean color dataset or a high-resolution model.

Finally, we do suspect that our results are dependent on the choice of ocean color product
resolution. We suspect that the major consequence is that $T_{L,\text{Chl}} \leq T_{E,\text{Chl}}$ for the reasons outlined
in our response to your comment (1).

We plan to include a more detailed discussion of why a 0.25º product was utilized and
specifically delineate what the limitations of this choice are and how it likely influences our
results (incorporating the last paragraph of our response to your comment (1)). In a revised
introduction we plan to clearly motivate an analysis of mesoscale variability as done here, and in
a revised conclusion we plan to recommend subsequent studies of submesoscale variability as
done here.

Changes made: We continued to work with the 0.25º GlobColour fields. We updated the text in
the following manners. We updated the last paragraph of the introduction (Section 1) to clarify
that this is a study of mesoscale variance and to motivate that choice of scale. We updated
Section 3.2.2 to clarify our choice of product. We wrote a new subsection of Results and
Discussions (4.6) where we discuss how our results are influenced by the choice of data products
and filtering (following the main points of our first Author Comment) and point out that an
analysis of submesoscale-resolving data may lead to different conclusions. We updated the
conclusions (Section 5) to reinforce that our results are indicative of mesoscale variance and to
suggest that future studies of submesoscale statistics are warranted.

3. The use of geostrophic velocities to estimate QPI is problematic. There are many processes
that attribute to Lagrangian decorrelation missing from these fields- I'm not even sure if Ekman
drift is included? Many of these forces are also likely to affect the upper ocean to larger extent,
creating an even further biasing when being omitted. One easy test is to calculate QPI for the
drifters the same way as the floats to see how representative the geostrophic velocity fields are.
Another option is to conduct the excercise in a high resolution ocean model using virtual drifters
and floats.

The QPI is calculated using trajectories computed from the global altimetry product, which includes a geostrophic term based on sea level anomalies and nothing else. The study of Della Penna et al. (2015) that developed the QPI compared distributions of QPI for SVP (real) drifters using trajectories calculated from different altimetry products (see their Supplementary Information Figure 4), including a global altimetry product (geostrophy only), a regional altimetry product (geostrophy only), and a regional Ekman corrected product (geostrophy + Ekman). Their conclusion was that "[using] different products does not alter significantly the shape and the extent of the [distribution of QPI], yet differences in the distributions can be observed in the tails". Though their study was performed in the Southern Ocean and though they compare trajectories in a slightly different manner than we do, we took this to mean a geostrophic term would likely dominate the trajectories, especially in the vicinity of the Gulf Stream and North Atlantic Current where a geostrophic balance is generally reasonable. Our choice was further motivated by our desire to study mesoscale variations in the velocity fields, which the altimetric geostrophic fields are known to capture reasonably well.

[Figure]

**Figure R1:** Probability mass functions (PMF) of QPI for all floats (top panel), floats with $\Delta t \approx 2$
days (middle panel), and drifters with $\Delta t = 2$ days (bottom). Vertical lines represent 5 km.

We computed the QPI for all of our drifter returns that fall in the study domain of [30N, 65N,
300E, 340E] and [2003-01-01, 2016-12-31] and used our daily subsampling, so that $\Delta t = 2$ days
since trajectories are compared at $[t_{i-1}, t_i, t_{i+1}]$. The distributions of drifter QPI are shown in
Figure R1 (third panel). We found two things surprising. Firstly, the median over all drifters is
larger than expected at about 8 km. Secondly, the distribution of float measured QPI when
restricting to profiles with $\Delta t \approx 2$ days is very similar (Figure R1, second panel), with only a
slightly larger median. Inspecting QPI as a function of latitude (averaged in 5° bins) for the two
platforms reveals that, while the distributions are similar when including all samples, the
latitudinal variations are different. Each platform has a maximum near the Gulf Stream (40-
45ºN) and a minimum at 50-55ºN or 55-60ºN, but the QPI is more variable for the floats, with a
maximum larger than that for drifters and a minimum that is smaller (even though, presumably,
floats are less Lagrangian with respect to the surface flow) (Figure R2). As we know that floats
tend to lag the surface flow, the energetic and sheared currents of the Gulf Stream may
exaggerate this difference, causing the very large QPI there. On the other hand, deeper mixed
layers and more sluggish currents at higher latitudes may cause a relatively smaller QPI for the
floats at higher latitudes. Another possibility is that Ekman transports become important farther
north away from the Gulf Stream, and the deeper floats are sheltered from this flow, instead
primarily feeling the geostrophic flow and yielding a relatively smaller QPI compared to drifters,
who feel the total current. However, given the relatively stable distribution of drifter QPI with
latitude, we suspect lack of including an Ekman term amounts to a small difference, in line with
the findings of Della Penna et al. (2015). Instead, deviations for the drifters are probably
primarily due to sub-map-grid scale processes or altimetric geostrophic currents generally
underestimating surface flow due to the finite differences being computed from a product that
has been mapped from the actual altimetry swaths (Ascani et al., 2013; Sudre and Morrow,
2008).

[Figure]

**Figure R2:** QPI as a function of latitude for drifters (black) and floats with $\Delta t \approx 2$ days (blue) in
5° latitude bins. Squares with vertical line are means ±1 standard error. Diamonds are medians.

All that said, we still think that the QPI as we calculate it (deviations from a surface geostrophic
trajectory) is reasonable for our study mainly because, as indicated (though we will clarify), we
have set out to conduct a study of mesoscale Eulerian-Lagrangian time and length scales. For
example, the velocity scale analysis (Figure 5) is based on Eulerian scales from satellite
altimetric geostrophic currents and drifter velocity time series filtered to remove super-inertial
variability. If we take that geostrophic altimetric fields are a reasonable approximation for the
mesoscale flow, then it is reasonable to us to emphasize float segments whose trajectories are
similar to trajectories subject to that flow.

Changes made: The above results from our Author Comment are summarized in a paragraph
added to Appendix A (the appendix that explains the QPI). The QPI is unchanged.

4. I'm not happy with how the Chl data for the floats is handled. The mean Chl concentration in
the mixed layer is not what is observed by satellite. This is of particular importance in regions
with deep Chl maxima where most Chl is close to the base of the mixed layer and not visible
from space. This issue can easily be amplified in this study if there is MLD variability over short
timescales or if the isolines are sloping. Each case could lead to spurious variability in Chl
observed by the float, compared to the drifters. The correct approach would be to use attenuation
or PAR from the float (or Kd490 from satellite in if not available on the float) and average the
Chl data down to the first optical depth. An even better approach would be to match satellite Chl
to the floats the same way as to the drifters. I don't see any benefits in using In-situ observations
for one platform and satellite-derived data for the other when comparing the two.

We have computed an alternate depth-reduced chlorophyll series from the floats that is meant to
better approximate what the satellites see. About 90% of the satellite-measured chlorophyll
signal in the open ocean comes from a depth of $1/Kd_{490}$, and it is exponentially weighted in the
vertical (Gordon and McCluney, 1975). Our new approach is to conduct a weighted average over
one attenuation depth. First, we estimate $Kd_{490}$ from the floats following Morel et al. (2007)
(their equation 8),

$$Kd_{490} = 0.0166 + 0.0773[\text{Chl}]^{0.6715},\tag{R1.1}$$

where we take [Chl] as the mixed-layer average chlorophyll. Then, we take a weighted vertical
average at each time step as

$$\text{Chl}_{\text{float}}(t) = \frac{\sum\limits_{z=1/Kd_{490}}^{z=\text{surface}} \text{Chl}(z,t)\exp\left(-2Kd_{490}(t)z\right)}{\sum\limits_{z=1/Kd_{490}}^{z=\text{surface}} \exp\left(-2Kd_{490}(t)z\right)}.\tag{R1.2}$$

We use a weighted sum instead of integral because some profiles have missing data near the
surface. The series is then log transformed and filtered as before. In general, the two time series
(R1.2 and the MLD-average used in the original manuscript) are very similar with some
discrepancies at daily to subdaily fluctuations; however, these are generally removed with the
subdaily filter (compare Figure B1 to attached revised Figure R7). We then reran all scales and
provide here a complete set of figures (equivalents to Figures 6-9, B1, D1). The results are not
appreciably different. We would be fine with using the new method of depth reduction.

As for using float-measured data instead of projecting satellite data onto float trajectories, we see
great value in using in-situ observations. Firstly, it is not possible to do this analysis with satellite data projected onto the floats, given the limited number of floats available. As you can see in
Figure B1 (squares), there are many gaps when projecting satellite ocean color onto a float
trajectory, even when using a nearest neighbor approach as is done in that figure, which is more
generous than the preferred bilinear interpolation used for drifters in the paper. This issue is not
prohibitive when working with the drifters because there is a tremendous number of them, so
even though individual segments are sparse and offer few pairs at a given lag, a composite ACF
can be constructed from many sparse segments. Secondly, we feel it is instructive to demonstrate
what can be learned from a near-continuous, in-situ time series, as this represents as complete a
dataset as is possible and is what most float users will work with.

[Figure]

**Figure R3:** Revised Figure 6 from manuscript using new definition of float chlorophyll series.

[Figure]

**Figure R4:** Revised Figure 7 from manuscript using new definition of float chlorophyll series.

[Figure]

**Figure R5:** Revised Figure 8 from manuscript using new definition of float chlorophyll series.

[Figure]

**Figure R6:** Revised Figure 9 from manuscript using new definition of float chlorophyll series.

[Figure]

**Figure R7:** Revised Figure B1 from manuscript using new definition of float chlorophyll series.

[Figure]

**Figure R8:** Revised Figure D1 from manuscript using new definition of float chlorophyll series.

Changes made: We continue to work with in-situ float data out of necessity. The depth-reduced
chlorophyll series for floats is now calculated using equations R1.1-R1.2 given in our Author
Comment (instead of the MLD-average) and all calculations and figures are revised. Section
3.2.1 is updated with a development and presentation of the equation (following our Author
Comment). New section 4.6 mentions that the results and conclusions are not sensitive to this
choice.

5. Finally, while the formalism in the MS is thorough and impressive, I think it might scare many
potentially readers away. Cleaning up the text by explaining the reasoning in a way that can be
understood by a wide audience and move a portion of the analytical description to an appendix
would probably increase the readership statistics and potential of citations.

This is a good suggestion for such a technical paper. Already, in our initial submission we've
made great effort to move non-essential information to appendices (there are already five). In our
reading, though there is a deep theoretical exposition in Section 2.1 and a lot of methodological
detail in Sections 3.3-3.5, we believe that the information retained in the main text is essential for
evaluation of the paper. However, we agree that the exposition can be improved. Referee #2
made some good suggestions on how to enhance the clarity of (and reduce length of) sections 3.3

and 3.4. We believe that by addressing those issues and by more clearly motivating our analyses,
readability will improve.

Changes made: Section 3.4 is shortened and much simpler. Section 3.3 was updated for clarity.
No new appendices were added.

**Author Comments in response to Referee #2**

This manuscript presents extensive work evaluating Eulerian and Lagrangian time and length
scales of velocity and chlorophyll, as well as discussion about how they correlate. The proper
interpretation of drifting phytoplankton observed in a Eulerian fashion is a longstanding
paradigm in ocean ecology. However, estimates of Lagrangian phytoplankton statistics and
comparisons with Eulerian counterparts are rare. This study represents a significant contribution
towards best understanding how to interpret phytoplankton/chlorophyll measured in both
Eulerian and Lagrangian platforms. The authors are very thorough in their analysis and
description of the results. Nonetheless, I have a few comments to be addressed prior to
recommending publication.

Thank you for your close read and evaluation of our manuscript.

Major comment: There are several data limitations that guide methodological decisions in an
analysis of this type (e.g., the broad spatial averaging, chlorophyll averaging in the MLD). While
some of the issues arising from these are mentioned briefly throughout the text, I would prefer to
see a dedicated discussion section with the limitations and caveats.

The averaging of scales (or compositing of ACFs) over [5º x 5º] space bins is meant to enhance
the quality of the estimates by averaging over a region that is relatively spatially homogenous.
Other authors doing a similar analysis of velocity in this region used [10º x 10º] space bins and
found this adequate to describe spatial variability in Lagrangian scales (Lumpkin et al., 2002).
We chose to use the same [5º x 5º] space bins as Glover et al. (2018), who calculated variograms
of satellite ocean color in each bin much like we compute ACFs and found these bins good to
resolve spatial variability. We will better motivate this in the text.

As for the depth reduction of chlorophyll, we had indeed used a simple average over the mixed
layer since other authors had done this and demonstrated good agreement with satellite ocean
color when describing seasonal variability in the region (Yang et al., 2020). At the suggestion of
Referee #1, we computed an alternate depth-reduced chlorophyll series from the floats that is
meant to better approximate what the satellites see. Please refer to our Author Comment to
Referee #1 for full details, but briefly, we utilize the fact that 90% of the satellite-measured
chlorophyll signal in the open ocean comes from a depth of $1/Kd490$, and it is exponentially-
weighted (Gordon and McCluney, 1975). We estimate $Kd_{490}$ from the floats following Morel et
al. (2007) (their equation 8),

$Kd_{490} = 0.0166 + 0.0773[\text{Chl}]^{0.6715}$

where we take [Chl] as the mixed-layer average chlorophyll, and then take a weighted vertical average at each time step as

$$\mathrm{Chl}_{\mathrm{float}}(t) = \frac{\sum\limits_{z=1/Kd_{490}}^{z=\mathrm{surface}} \mathrm{Chl}(z,t)\exp\left(-2Kd_{490}(t)z\right)}{\sum\limits_{z=1/Kd_{490}}^{z=\mathrm{surface}} \exp\left(-2Kd_{490}(t)z\right)} \; .$$

The series is then log transformed and filtered as before. After rerunning all scales (please refer
to set of figures in Author Comment to Referee #1), the results are not appreciably different. We
will include a description of this comparison in the text.

As for the choice of ACF parameters in Table 1, please refer to our response to your comment
below.

We plan to consolidate all of the above matters (spatial averaging, depth averaging, ACF
parameters) into a subsection of the Discussions, as you suggest.

Changes made: At your suggestion, we wrote a new Section 4.6 (Methodological decisions) that
includes all the issues mentioned in the Author Comment and how they might influence our
results: depth-reduction of float data, ACF parameters and spatial averaging, choice of ocean
colour product and filtering.

Specific comments:

I find that, while technically correct, talking about Lagrangian-Eulerian "statistics" in the title
and throughout the text can be misleading. Why not refer to the specific statistics that are
included in the analysis? i.e., Lagrangian-Eulerian time and length scales.

We felt that use of the word "statistics" made for a more compact title, with the meaning
becoming clear after reading the abstract. But we do not object to changing the title to:
"Lagrangian-Eulerian time and length scales of mesoscale ocean chlorophyll from Bio-Argo
floats and satellites".

Changes made: We suggest that the title be changed (substituting "time and length scales" for
"statistics") if the Editor allows. On a few occasions, we continue to use "statistics" for brevity,
noting that the first sentences of the Introduction (Section 1) and Conclusions (Section 5) define
our use of "statistics" as "time and length scales".

The notation of upper case L for both Lagrangian and length-scale can be a bit confusing. I
suggest using upper and lower case or a different notation to improve readability.

We agree about the confusion. We thought about using upper and lower case letters but this can
become problematic since a lower case "l" can look like a number 1 or capital "I" or something
else. We propose to maintain "T" and "L" for scales and replace subscripts "L" and "E" with
either "l" and "e" or "LAG" and "EUL" for Lagrangian and Eulerian, respectively, depending on
which of the two looks best.

Changes made: All subscripts "E" for Eulerian and "L" for Lagrangian are changed to "e" and
"l", respectively, in all text, equations, and figures.

Equation 5. Terminology becomes confusing here too when calling the nominators Lagrangian,
Eulerian and Spatial (chlorophyll) scales. Is there a different name that could be more
appropriate and less confusing? This is essentially a change in chlorophyll, correct?

These are effectively standard deviations of chlorophyll computed in different frames: from
Lagrangian time series (subscript "*LAG*" or "*l*") from Eulerian time series (subscript "*EUL*" or
"*e*"), or from spatial maps (subscript "spatial"). Though less than satisfactory, we cannot think of
a better notation to express this point. However, we could add to the text the literal definition of
each term as supplied here in our Author's Comment document.

Changes made: We could not find a better terminology here. To help, we now indicate in the text
that angle brackets indicate standard deviations, and we note that these terms are defined in
Table 1.

Table 1 is also confusing. Why are ACF bins different? Why are time windows for Eulerian and
Lagrangian different? Does that have any effect on the comparison? (I think it would if you were
calculating other statistics). Where does the 27.8km ACF bin for Eulerian length scale come
from? I probably missed it.

As indicated in your Major Comment above, we plan to better motivate these choices and
consolidate them into a subsection of Discussions that will cover all methodological choices.
Briefly, we address your specific questions here.

Regarding temporal ACF bin sizes: Ideally, one would use a bin size that matches the sampling
interval of the time series because this is the smallest lag that can be resolved. For this reason,
the ACFs based on satellite altimetry, satellite ocean color, or drifters use a bin size of 1 day. The
floats have a variable profiling interval. While they sometimes profile with a frequency of about
1 per day, they generally profile less frequently and we found a bin size of 5 days to be a
reasonable choice (with smaller bin sizes, many segments would offer no pairs). As we state, the
two metbio* float segments are given special attention because they profile more frequently, and
for that reason we were able to use a finer bin size of 1 day. As a general statement, choosing a
larger bin size for the ACF causes structure (curvature) of the ACF to be poorly resolved at short
lag and biases time scales large. This point is brought up in section 4.2.

Regarding temporal segment lengths for ACF analysis:  The Lagrangian segments should be kept
as short as possible because as a platform moves it may encounter different environmental
(physical or otherwise) conditions, and we found 120 days was a reasonable length of time. For
Eulerian segments, this is not an issue, and, since seasonal variability is removed, length of the
segment is generally unimportant. Given that, we used 365-366 day segments for chlorophyll out
of convenience since the data were stored as yearly files.

Regarding spatial ACF bin size: Related to our point about temporal ACF bin sizes, it would be
best to use a bin size equal to the data spacing. We chose 27.8 km as that approximately
corresponds to the 0.25º resolution of the data in the latitudinal direction. Obviously, pairs
spaced zonally may have a separation less than that distance and would fall into the first bin.

Changes made: The above discussion from our Author Comment is included in our new Section
4.6 (Methodological decisions).

Figure 1d. orange profiles: QPI<5km; blue: all others (i.e., not total)

You are correct. The total height of the bars represents the total number of profiles, but the blue
region represents only the portion with QPI > 5 km. We will fix this.

Changes made: The caption of Figure 1 has been corrected accordingly.

Line 180. Please specify the convention for flagging. It is my understanding that BGCArgo
flagging may have changed through the years and between institutions. (I've used Sprofs where
3 means bad).

We will provide a brief description here and include reference to the relevant Argo user manual
for more information.

Changes made: The flag levels have been defined and a citation to Argo Data Management Team
(2019) added.

How does the GlobColour product compare to other products? Why is this one selected over
others? (OCCCI, for instance). I suggest including a brief sentence.

We did not compare how different satellite products affect the scales that are calculated. We
chose GlobColour because it is blended from all available satellites and is therefore probably a
most complete product in terms of space-time coverage without interpolation. Further, the study
of Zhang et al. (2019) demonstrated that GlobColour data projected onto surface drifter tracks
resolve realistic Lagrangian behavior in terms of (sub)mesoscale dynamics, so we conclude that
their space-time information is biophysically accurate. We plan to include this information in the
manuscript, in the Methods section where we introduce the data. As for why we chose to use a
25 km product, that requires a more nuanced discussion and we refer you to please see our
Author Comment to Referee #1. We propose to include that discussion in a revised Discussions
section.

Changes made: The motivations for using a 0.25° ocean colour product, the rationale for why
GlobColour specifically was chosen, and how the choice of this product may influence our
results are all discussed in the new Section 4.6 as it seemed to fit better there.

Section 3.3 could be simplified. Two sets of chlorophyll anomalies are estimated: 1. Anomalies
with respect to a 31-day smoothing filter, and 2. Anomalies with respect to the climatology. I
would suggest stating something like that to start, and then continue with the details.

This is a reasonable suggestion, and we can modify the opening sentence of section 3.3
accordingly.

Changes made: The opening paragraph of Section 3.3 has been updated following your
suggestion.

The climatology is based on the same 31-day filter + a boxcar function? This is not exactly what
comes to my mind when "climatology" or "repeating annual cycle" is mentioned.

We apologize for the confusion here. This is an admittedly technical point so we left the details
in the Appendix B, but perhaps we need to clarify the main text. Essentially, the "smoothed"
subtrahend is from a 3-D convolution with a filter kernel that is a 2-D Gaussian in space and a
31-day Hamming window in time. The "climatology" subtrahend comes from first stacking the
arrays by day of year in a 4th dimension so that the convolution is with a 4-D kernel that is a 2-D
Gaussian in space, a 31-day Hamming window in day-of-year (like a Julian day, not absolute calendar date), and a boxcar (equal weights) across years. That way, as we say in the Appendix
B, "[for] example, January 1 of every year is regarded as having the same time coordinate". The
end result is a set of maps for each day-of-year, hence making it a repeating annual cycle. We
can move the illustrative sentence (reproduced here) to the main text for clarity.

Changes made: The third paragraph of Section 3.3 has been updated to clarify how the
"climatology" subtrahend is constructed. We follow the outline given in our Author Comment
and use text from Appendix B.

I don't like the use of the satellite-based "subtrahend" to estimate chlorophyll anomalies from the
MLD-averaged chlorophyll from the float. How does the MLD average compare to the satellite?
I think some type of bias correction may be needed. You mention that the subtrahend is
regressed against float data. Do you mean you corrected a bias? That should be included in a
supplement.

It is not possible to construct a "climatology" subtrahend from the floats because there is not
enough interannual coverage of floats over the spatial footprint of the horizontal component of
the filter at any given time step. For this reason, we need to turn to climatological fields
constructed from the satellite data. To illustrate how the subtrahends look (and how float and
satellite data compare), we included Figure B1. We think that figure illustrates that the satellite-
constructed "climatology" subtrahend is reasonable to compare with the float data. As you point
out, the regression effectively serves as a bias correction so that the mean and range of the
subtrahend (once it is projected onto the floats) is comparable to the mean and range of the float-
measured chlorophyll. The details of the procedure are described in the existing Appendix B. If it
is helpful, regression coefficients can be included.

Changes made: We continue to use the satellite data to construct the "climatology" subtrahend
for the floats. We updated Appendix B to refer to the regression as serving like a "bias
correction" and include the equation.

Line 240. Why aren't Eulerian and Lagrangian segments equal?

This is a matter of convenience. The Lagrangian segments should be kept as short as possible
because as a platform moves it may encounter different environmental (physical or otherwise)
conditions, and we found 120 days was a reasonable length of time. However, this is not
important for Eulerian segments, especially since low frequency (such as seasonal) variability
has been removed. Since the Eulerian data are stored in annual files, it was easiest to work with
year-long segments.

Changes made: This answer is given in the new Section 4.6 (Methodological decisions).

Section 3.4 could be simplified as well. If I understand correctly, you tested two approaches to
estimate spatially averaged scales. In lines 272-275 you mention you use one or the other. When
and why you use each one should be clearer.

This is basically correct. When possible, we apply both methods (e.g., compare Figures 6 and
D1), but only equation 8 is an option for any scales derived from ocean color due to the large
number of gaps. In simplest terms: "All scales are derived by averaging in space (from
integrating Eq. (7) and averaging), except any scales involving satellite ocean color, where large
numbers of gaps require computing scales from space-composited ACFs (from integrating Eq.

(8)).” We can open the discussion on lines 272-281 with the preceding simple sentence and then
eliminate much of the redundant (and less clear) text that follows.

Changes made: Section 3.4 has been rewritten following our Author Comment above.

Line 292. Picks?

Sorry: “picks” should read “scales”.

Changes made: The typo has been corrected.

Lines 319-320: “If we take …” this sentence is confusing.

We apologize for the confusion here. Our intention is to draw some contrast between float
profiles where the QPI is “small” and “large”. While the distribution in Figure 2 is continuous
and there is no real threshold, we noted that there is a mode of profiles between zero and 5 km,
so we chose this threshold for display purposes. As we mention in the text, 5 km is a good
compromise between having a large amount of profiles and having a QPI that is small, so it
serves as a reasonable threshold between a “small” and “large” QPI for the purposes of display in
Figures 2-3. Other than for display purposes in those figures, though, QPI is only used for
weighting averaged scales and there is no use of a threshold in Figures 4-9 or their
interpretations. We can update the text with the information supplied in this Author’s Comment
document to clarify where the 5 km threshold comes from and when and why it is used.

Changes made: The opening paragraph of Section 4.1 has been rewritten following the outline
given in our Author Comment above to make it clear that the threshold of 5 km is arbitrary and
for display purposes only.

I probably missed this. Are the results in figures 5 to 9 based on all profiles or only QPI<5km?

We apologize for the confusion here. Figures 5 to 9 display results based on all float profiles. We
can update the captions to convey this. The filled circles treat all float segments equally in the
averages whereas the crosses weight by segment-median QPI$^{-2}$ so that segments with smaller
QPI count more.

Changes made: The caption in Figure 5 has been updated. In our reading, the edits to Section 4.1
now make it clear that the threshold of 5 km is for display purposes only, and that the threshold
has no bearing on Figures 5-9 or their discussion.

**References:**

Argo Data Management Team: Argo user’s manual, Ifremer, https://doi.org/10.13155/29825,
2019.

Ascani, F., Richards, K. J., Firing, E., Grant, S., Johnson, K. S., Jia, Y., Lukas, R., and Karl, D.
M.: Physical and biological controls of nitrate concentrations in the upper subtropical North
Pacific Ocean, Deep Sea Res. Part II Top. Stud. Oceanogr., 93, 119–134,
https://doi.org/10.1016/j.dsr2.2013.01.034, 2013.

Della Penna, A., De Monte, S., Kestenare, E., Guinet, C., and d’Ovidio, F.: Quasi-planktonic
behavior of foraging top marine predators, Sci. Rep., 5, 18063,
https://doi.org/10.1038/srep18063, 2015.

Eveleth, R., Glover, D. M., Long, M. C., Lima, I. D., Chase, A. P., and Doney, S. C.: Assessing
the Skill of a High-Resolution Marine Biophysical Model Using Geostatistical Analysis of
Mesoscale Ocean Chlorophyll Variability From Field Observations and Remote Sensing, Front.
Mar. Sci., 8, 1–10, https://doi.org/10.3389/fmars.2021.612764, 2021.

Glover, D. M., Doney, S. C., Oestreich, W. K., and Tullo, A. W.: Geostatistical analysis of
mesoscale spatial variability and error in SeaWiFS and MODIS/Aqua global ocean color data, J.
Geophys. Res. Oceans, 123, 22–39, https://doi.org/10.1002/2017JC013023, 2018.

Gordon, H. and McCluney, W.: Estimation of the depth of sunlight penetration in the sea for
remote sensing, Appl. Opt., 14, 413–416, https://doi.org/10.1364/AO.14.000413, 1975.

Lumpkin, R., Treguier, A.-M., and Speer, K.: Lagrangian eddy scales in the Northern Atlantic
Ocean, J. Phys. Oceanogr., 32, 2425–2440, 2002.

Morel, A., Hout, Y., Gentili, B., Werdell, P. J., Hooker, S. B., and Franz, B. A.: Examining the
consistency of products derived from various ocean color sensors in open ocean (Case 1) waters
in the perspective of a multi-sensor approach, Remote Sens. Environ., 111, 69–88,
https://doi.org/10.1016/j.rse.2007.03.012, 2007.

Sudre, J. and Morrow, R. A.: Global surface currents: a high-resolution product for investigating
ocean dynamics, Ocean Dyn., 58, https://doi.org/10.1007/s10236-008-0134-9, 2008.

Yang, B., Boss, E. S., Haëntjens, N., Long, M. C., Behrenfeld, M. J., Eveleth, R., and Doney, S.
C.: Controls on the North Atlantic Phytoplankton Bloom: Insights from Profiling Float
Measurements, Front. Mar. Sci., 7, 139, https://doi.org/10.3389/fmars.2020.00139, 2020.

Zhang, Z., Qiu, B., Klein, P., and Travis, S.: The influence of geostrophic strain on oceanic
ageostrophic motion and surface chlorophyll, Nat. Commun., 10, 1–11,
https://doi.org/10.1038/s41467-019-10883-w, 2019.

---

## Referee Report (RR1)

Thank you for your response to my comments and questions. I think there is a nice study hiding in the manuscript, but that there is still some work to find it.

The study addresses important questions and suggests an interesting framework for comparing Lagrangian and Eulerian scales, but while I respect the intention by the authors to only include mesoscales, I think this constraint has to be communicated more clearly. I originally assumed that the main story was to assess if Argo floats can be assumed Lagrangian when sampling Chl, but such analysis would need to include all scales that can be observed. I now realize that this assessment is of lower priority and that you mainly focus on understanding how Eulerian and Lagrangian timescales compare over mesoscales. This focus is of course valid, but the abstract, introduction ,and conclusions should be rewritten to deemphasize the question about the utility of Argo floats and if they can be considered Lagrangian. Also, please be careful when providing estimates of timescales of decorrelation since these are calculated for a simplified world without sub-mesoscale processes.

I am still quite concerned about your definition of material derivatives and the consequence it has on your results. Eq 1, as it is stated now, is correct when describing the material derivative of a field which is fixed in space, for example the temperature gradient in a small lake (https://en.wikipedia.org/wiki/Material_derivative) or a stationary velocity field as used by Middleton (1985). I don't think it's correct for Chl in the open ocean which will be advected together with the Lagrangian reference point though. Here, the material derivative in a Lagrangian frame is

$$\frac{DChl}{Dt} = \frac{\delta Chl}{\delta t}$$

And in a Eulerian frame

$$\frac{DChl}{Dt} = \frac{\delta Chl}{\delta t} + \boldsymbol{u}\nabla Chl$$

Please see for example eqs 1 and 2 in Chenillat 2015 (https://www.frontiersin.org/articles/10.3389/fenvs.2015.00043/full) or section 1.2.2 in https://www.usc.es/export9/sites/webinstitucional/en/investigacion/grupos/gfnl/documents/thesis/tesis_Florian.pdf. The paragraph on lines 85-97 is a bit confusing due to this. I read it as starting with talking about Chl, making a statement based on the material derivative of velocity in the middle, switching back to to talk about Chl, and finishing with a relationship based on Lagrangian and Eulerian observations of velocities. I am a bit reluctant to take the rest of the section at face value, especially equations 4 and 5, due to this. It might be that you can expand the findings by Middleton (1985) to a moving tracer, which is different from their assumptions of stationarity, but it would have to be carefully proven.

Finally, I still think that the organization and tone of the MS miss the intended audience. For example, I would have liked a more verbose discussion about the formalism for relating Eulerian and Lagrangian timescales described in Middleton (1985) and why it can be used for contrasting them. I'm also missing a more descriptive explanation of the different metrics that being used. What does for example $u'/c^*_{Chl}$, $\alpha_{Chl}$, or $q_{Chl}$ tell us? It can be found by reading the text and references carefully, but a reader might give up before figuring it out. Just a table listing all parameters and a short description for each of them would be very helpful. The description of ACF is very thorough but it's not easy to figure out what is specific with your approach without going through the section in detail. Finally, it would be good to add references to the equations that aren't original to this MS.

---

## Author Response (AR2)

Lagrangian-Eulerian time and length scales of mesoscale ocean chlorophyll
from Bio-Argo floats and satellites

Darren C. McKee[1], Scott C. Doney[1], Alice Della Penna[2,3], Emmanuel S. Boss[4], Peter Gaube[5],
Michael J. Behrenfeld[6], David M. Glover[7]

[1]Department of Environmental Sciences, University of Virginia, Charlottesville, VA, 22904,
USA
[2]Institute of Marine Science, University of Auckland, Auckland, New Zealand
[3]School of Biological Sciences, University of Auckland, Auckland, New Zealand
[4]School of Marine Sciences, University of Maine, Orono, ME, USA
[5]Applied Physics Laboratory, University of Washington, Seattle, WA, USA
[6]Department of Botany and Plant Pathology, Oregon State University, Corvallis, OR, USA
[7]Department of Marine Chemistry and Geochemistry, Woods Hole Oceanographic Institution,
Woods Hole, MA, USA

**Format of this document:**
Black = Referee Comments; Blue = Author Comments and description of how manuscript has
changed.

**Author Comments in response to Referee #1**

Thank you for your response to my comments and questions. I think there is a nice study hiding
in the manuscript, but that there is still some work to find it.

Thank you for your careful review of our revised manuscript.

The study addresses important questions and suggests an interesting framework for comparing
Lagrangian and Eulerian scales, but while I respect the intention by the authors to only include
mesoscales, I think this constraint has to be communicated more clearly. I originally assumed
that the main story was to assess if Argo floats can be assumed Lagrangian when sampling Chl,
but such analysis would need to include all scales that can be observed. I now realize that this
assessment is of lower priority and that you mainly focus on understanding how Eulerian and
Lagrangian timescales compare over mesoscales. This focus is of course valid, but the abstract,
introduction ,and conclusions should be rewritten to deemphasize the question about the utility of
Argo floats and if they can be considered Lagrangian. Also, please be careful when providing
estimates of timescales of decorrelation since these are calculated for a simplified world without
sub-mesoscale processes.

That is a correct assessment of our intention: we primarily aim to understand how Lagrangian
Chl scales relate to Eulerian Chl scales, and how the velocity field provides the link. Judgment of
floats' suitability in Lagrangian analysis is a secondary aim. We have revised the text to de-
emphasize an assessment of the suitability of profiling floats in Lagrangian analysis through the
following changes: (1) We removed the last two paragraphs of the conclusions; (2) We removed
the last sentence of the abstract and replaced it with a comment on the importance of stirring for
setting Lagrangian scales, which is a conclusion related to our primary aim. Nevertheless,
because we use floats as a tool, an assessment of their behavior is necessary, as is some introductory description of how they sample. Therefore, we haven't changed the introduction or
other sections.

Secondly, our approach revolves around calculation and interpretation of scales computed from
data. However, the abstract, introduction, and conclusions (even the title) indicate that they are
computed from mesoscale-resolving (or filtered) data. Therefore, we prefer to refer to scales as
simply "integral" or "decorrelation" scales. Discussions (Sect. 4.2-4.6) and Conclusions (Sect. 5)
are clearly framed in terms of mesoscale processes and clearly discuss implications of analyzing
mesoscale-resolving data (or equivalently, data that do not resolve the submesoscale).

I am still quite concerned about your definition of material derivatives and the consequence it
has on your results. Eq 1, as it is stated now, is correct when describing the material derivative of
a field which is fixed in space, for example the temperature gradient in a small lake
(https://en.wikipedia.org/wiki/Material_derivative) or a stationary velocity field as used by
Middleton (1985). I don't think it's correct for Chl in the open ocean which will be advected
together with the Lagrangian reference point though. Here, the material derivative in a
Lagrangian frame is
$\frac{D Chl}{D t} = \frac{\delta Chl}{\delta t}$
And in a Eulerian frame
$\frac{D Chl}{D t} = \frac{\delta Chl}{\delta t} + \boldsymbol{u} \nabla Chl$
Please see for example eqs 1 and 2 in Chenillat 2015
(https://www.frontiersin.org/articles/10.3389/fenvs.2015.00043/full) or section 1.2.2 in
https://www.usc.es/export9/sites/webinstitucional/en/investigacion/grupos/gfnl/documents/thesis
/tesis_Florian.pdf. The paragraph on lines 85-97 is a bit confusing due to this. I read it as starting
with talking about Chl, making a statement based on the material derivative of velocity in the
middle, switching back to to talk about Chl, and finishing with a relationship based on
Lagrangian and Eulerian observations of velocities. I am a bit reluctant to take the rest of the
section at face value, especially equations 4 and 5, due to this. It might be that you can expand
the findings by Middleton (1985) to a moving tracer, which is different from their assumptions of
stationarity, but it would have to be carefully proven.

Thank you for suggesting the Chenillat et al. (2015) reference. We have perused the reference
and thought carefully about your comments. Our equation (1) makes no assumption about a
steady Chl field: we assume that it is fully evolving and advected by the velocity field (and also
subject to sources, sinks, and diffusion). Also, Middleton (1985) does not assume the velocity
field is steady, it is only assumed statistically stationary. Our equation (1) is the standard material
derivative in any text, and it is also the exact equation (1) of Chenillat et al. (2015), though they
refer to it as an advection-diffusion equation. When relating frames, the equation (2) in Chennilat
et al. (2015) groups the Eulerian (EUL) and advection (ADV) terms from equation (1) into what
they call an "evolution equation along a moving fluid parcel" (note the different notation in the
derivative of their equation (2), now using the *d* instead of partial $\partial$), the same as our term
"LAG" in our equation (1).

The use of our manuscript's equation (1) is standard, and our attribution of the three terms from
left to right as "LAG", "EUL", and "ADV" is also standard. For example, please refer to Jönsson
et al. (2011) and their equation (1), which is also the same as our equation (1). In their
manuscript, the left-hand side term (same as our LAG) is treated as a time derivative along a
trajectory, the fixed-in-space partial time derivative term (same as our EUL) is treated as a time derivative at a fixed location, and the advection term (same as our ADV) is taken as the
difference of the two. This is entirely consistent with our approach, the only difference being we
apply a statistical scaling of the terms (our equation (5)) in our study instead of quantifying the
terms with numerical data as those authors did. Both studies consider a fully evolving tracer
field. Therefore, we firmly believe that the mathematical formulation we employ is sound.

Regarding your comments about lines 85-97, here we explain why we first scale equation (1)
with velocity scales, and then with tracer scales. The analyses of Philip (1967) and Middleton
(1985) assume a velocity field that is statistically stationary and show that the ratio of velocity
time scales $T_l/T_e$ is a function of the ratio of velocity fluctuations $u'$ to an evolution speed of
the velocity field, $c^* = L_e/T_e$ . Their relationships are determined from velocity autocorrelation
functions and contain no information about tracer concentrations. Therefore, the dispersion (with
coefficient $K$) implied from the velocity scales – which is related to $T_l$ by $K = T_l(u')^2$ – is a
particle dispersion, representing effects of chaotic advection on the movement of water parcels.
Tracers like Chl are not particles and are subject to a transport that also includes diffusion (and
sources) in addition to advection. By scaling equation (1) with Chl scales (as done in equations
(4)-(5)), our intention is to see how velocity fluctuations $u'$ relative to translation of the Chl field
(at speed $c^*_{Chl} = L_{e,Chl}/T_{e,Chl}$ ) influence the values of $T_{l,Chl}$ and $T_{e,Chl}$. This approach uses scales
that have the effects of transport and non-conservative terms built in. Our goal is to gain insight
into how $T_{l,Chl}$ varies, and what processes control it. We do not know ahead of time if there will
be such a relationship as our equation (4), but given the earlier studies that conclude mesoscale
Chl anomalies can largely be explained by stirring (Denman and Abbott, 1988; Glover et al.,
2018), we suspect it is worth evaluating. We have reworded parts of Sect. 2 to make our logic
clearer, following the discussion above.

Finally, we hope that a clarification of our physical interpretation of the primary results will
alleviate your concerns about our framework and convince you that the Chl field is fully
evolving. We suspect that you might be objecting to our claim that advection can be important
for Lagrangian decorrelation. Indeed, the flow field is advecting the Chl so it might be surprising
for the Chl concentration along a trajectory to be influenced by advection as opposed to non-
conservative terms from the right-hand side of equation (1). Since LAG = EUL + ADV = S +
DIFF, even though the "real" drivers of Chl decorrelation along a trajectory must be sources /
sinks (S) or effects of turbulent diffusion (DIFF), mathematically, those terms project onto the
EUL and ADV terms.

To help clarify this point, we have made changes to the text to explain the origin of Lagrangian
decorrelation, suggesting mesoscale stirring is a major driver. The biggest change is an update to
Sect. 4.5. In earlier versions of the manuscript, this section simply presented the empirical
relation (our Eqns. (4) and (12), displayed in Figure 6) to interpret our results but now we use it
as an opportunity to tie all results together. We first update the section with a qualitative
interpretation of the functional form and its parameters (as you asked for in a later comment) and
we better motivate the idea that mesoscale stirring generates Chl anomalies and their Lagrangian
scales. This is done in part by appealing to the mixing length arguments of Glover et al. (2018),
who construct an additional scale (their $L_{tracer}$) equal to the distance a mesoscale eddy could stir a
water parcel containing Chl anomalies (their equation (2)) assuming that all Chl anomalies are generated by stirring a mean gradient. They show that in our study region of the North Atlantic,
$L_{\text{tracer}}$ and $L_{e,\text{Chl}}$ are statistically equivalent (see their Figure 7). This relation implies that the
statistical decorrelation length of Chl, $L_{e,\text{Chl}}$, is likely set by mesoscale stirring of the Chl field,
consistent with our finding that $L_{e,\text{Chl}} \approx L_e$, which is the velocity decorrelation scale and the
diameter of typical mesoscale eddies. The "geometry" of mesoscale stirring relates the frames
(Euclidean statistical separation $L_{e,\text{Chl}}$ and trajectory distance $L_{l,\text{Chl}}$) in the limit of large turbulent
velocity $u'$ by setting $q_{\text{Chl}}$. We suggest it is useful to think of ADV as a local stirring of the
mean Chl field as opposed to advection of anomalies over long distances (though we show below
in this response document that the two views are equivalent). That is why it matters in its relation
to the translation of the Chl field, given by $c_{\text{Chl}}^* = L_{e,\text{Chl}}/T_{e,\text{Chl}}$. In addition, it is best to view $u'$ as
turbulent velocity fluctuations. When the observer is a true surface Lagrangian observer (and the
velocity field is unfiltered with all velocity scales resolved), $u'$ is properly captured by the
platform's movements, but for an observer like an Argo float, effects of stirring are
underestimated. That means the ADV term is underestimated as we had said originally in the
manuscript.

This leads us to address what processes cause the decorrelation of Chl along a trajectory. We
make an important update to equation (1) to include a term DIFF, which encompasses effects of
turbulent diffusion due to unresolved advection, which nominally is due to the fact that water
parcel trajectories differ from infinitesimally small tracer particle trajectories but is even more
important by our focus on mesoscale variance since a range of scales of advection are not
resolved. Then, we continue our discussion in Sect. 4.5 where we introduce a scaling DIFF (new
equations 13-14) and consider the ratio of the LAG and DIFF terms (β = LAG/DIFF),
quantifying how much of the Lagrangian tendency is caused by turbulent diffusion (or
unresolved advection; equation 15). We can show that when turbulent velocity fluctuations are
relatively important ($u' > c_{\text{Chl}}^*$), LAG is largely explained by DIFF, a finding consistent with our
interpretation of stirring playing a leading role in setting Lagrangian statistics. Likewise, through
an inequality on β we are able to infer that sources ($S$ in equation 1) must be increasingly
important when turbulent velocity fluctuations are relatively small ($u' < c_{\text{Chl}}^*$). Noting this, in Sect.
4.4, we de-emphasize an attribution of biological sources and sinks S in driving LAG, noting that
in general DIFF could also be important. Finally, throughout we relax language that advection
"causes" Lagrangian decorrelation.

As an aside, though we do not include this in the manuscript to avoid the complications of
introducing an additional length scale, if we take the Glover et al. (2018) definition
$L_{\text{tracer}} = \langle \text{Chl} \rangle_{\text{space}} / \nabla \overline{\text{Chl}}$ (their equation 2) and take their finding that $L_{e,\text{Chl}} \propto L_{\text{tracer}}$ over our study
domain (their Figure 7), then our scaling $\text{ADV} = u' \langle \text{Chl} \rangle_{\text{space}} / L_{e,\text{Chl}}$ (our equation 5) becomes
$\text{ADV} = u' \nabla \overline{\text{Chl}}$. This supports our interpretation of ADV as local stirring of a mean gradient.

Finally, I still think that the organization and tone of the MS miss the intended audience. For
example, I would have liked a more verbose discussion about the formalism for relating Eulerian
and Lagrangian timescales described in Middleton (1985) and why it can be used for contrasting
them. I'm also missing a more descriptive explanation of the different metrics that being used.
What does for example $u'/c_{\text{Chl}}^*$, $\alpha_{\text{Chl}}$, or $q_{\text{Chl}}$ tell us? It can be found by reading the text and references carefully, but a reader might give up before figuring it out.
Just a table listing all parameters and a short description for each of them would be very helpful.
The description of ACF is very thorough but it's not easy to figure out what is specific with your
approach without going through the section in detail. Finally, it would be good to add references
to the equations that aren't original to this MS.

There is now additional detail in Sect. (2) about the interpretation of Middleton (1985) as a
"particle dispersion", motivating our use of tracer time and length scales, but just as the methods
section is already technical, we don't think a discussion of the formalisms leading to equation (3)
would benefit our audience. The parameters $u'$, $c^*_{\mathrm{Chl}}$, and $\alpha_{\mathrm{Chl}}$ are defined in Sect. 2, and $q_{\mathrm{Chl}}$ is
defined in Sect. 4.5. All of those parameters are given a qualitative interpretation in Sect. 4.5,
where we clarify in detail the physical processes leading to the $T_{l,\mathrm{Chl}}/T_{e,\mathrm{Chl}}$.

Finally, we have checked equations for attribution. Equation (1) is standard, but given its central
role we have added citations to Chennilat et al. (2015), Jönsson et al. (2011), d'Ovidio et al.
(2013), and van Sebille et al. (2018). Equation (3) is attributed to Middleton (1985). Equation
(6a) is attributed to Morel et al. (2007). We add references to Glover et al. (2011, 2018) for
equations (C1)-(C2). All other equations either follow from the above or are standard definitions.

**Author Comments in response to Referee #2**

This manuscript presents extensive work evaluating Eulerian and Lagrangian time and length
scales of velocity and chlorophyll, as well as discussion about how they correlate. As mentioned
by the authors, there is a lack of studies [at all scales of variability] comparing estimates of
Lagrangian and Eulerian phytoplankton statistics, including temporal and spatial correlation
scales at all scales. In this sense, this study represents a significant contribution towards best
understanding phytoplankton/chlorophyll measured in both Eulerian and Lagrangian platforms.
Throughout the revisions the authors have made an effort to improve the readability of a
technically loaded manuscript. I recommend this manuscript for publication after the following
details are considered:

Thank you for your careful evaluation of our revised manuscript.

1. Lines 77-81. I see these lines are responding to reviewer 1's comments, but this sentence is
very long, and the key points may be missed. Please consider rewording and breaking it up into
smaller sentences.

The sentence has been split and rewritten for clarity.

2. Line 220. Typo: "is used".

We adjusted this sentence by removing "used to calculate integral scales", since that point is
obvious. That simplification seems to be the best way to fix the sentence.

3. Lines 588 – 590, regarding temporal segment lengths for ACF analysis. Please consider
rewording. While I agree that given the range of temporal scales of phytoplankton (1 – 15 days),
the segment length of Eulerian time series is probably not an issue, the initial date may have an
impact in the result. If I understand correctly, this initial date is variable in the in situ data, but the same for every year in the satellite data. I'm not certain either what the effect would be in the
spatial scale estimate. Also consider that methods to remove seasonal variability are not perfect,
and some signal may still remain. I respect your methodology, but I don't think you are showing
evidence to definitely say that the different length segments for L and E estimates are "not an
issue" and "generally unimportant". I would be more cautious in this statement. In my own
experience, length of the time series did matter in the comparison of other Eulerian and
Lagrangian chlorophyll statistics. The storage format of the is not a strong reason for this
methodological choice, so I suggest removing that last part of the sentence.

Thank you for your concern and for sharing your experience. We have rewritten the sentence in
line with your suggestions.

**References:**

Chenillat, F., Blanke, B., Grima, N., Franks, P. J. S., Capet, X., and Rivière, P.: Quantifying
tracer dynamics in moving fluids: a combined Eulerian-Lagrangian approach, Front. Environ.
Sci., 3, https://doi.org/10.3389/fenvs.2015.00043, 2015.

Denman, K. L. and Abbott, M. R.: Time evolution of surface chlorophyll patterns from cross-
spectrum analysis of satellite color images, J. Geophys. Res., 93, 6789–6798,
https://doi.org/10.1029/JC093iC06p06789, 1988.

Glover, D. M., Jenkins, W. J., and Doney, S. C.: Modeling Methods for Marine Science,
Cambridge University Press, Cambridge, UK, 592 pp., 2011.

Glover, D. M., Doney, S. C., Oestreich, W. K., and Tullo, A. W.: Geostatistical analysis of
mesoscale spatial variability and error in SeaWiFS and MODIS/Aqua global ocean color data, J.
Geophys. Res. Oceans, 123, 22–39, https://doi.org/10.1002/2017JC013023, 2018.

Jönsson, B. F., Salisbury, J. E., and Mahadevan, A.: Large variability in continental shelf
production of phytoplankton carbon revealed by satellite, Biogeosciences, 8, 1213–1223,
https://doi.org/10.5194/bg-8-1213-2011, 2011.

Middleton, J. F.: Drifter spectra and diffusivities, J. Mar. Res., 43, 37–55, 1985.

Morel, A., Hout, Y., Gentili, B., Werdell, P. J., Hooker, S. B., and Franz, B. A.: Examining the
consistency of products derived from various ocean color sensors in open ocean (Case 1) waters
in the perspective of a multi-sensor approach, Remote Sens. Environ., 111, 69–88,
https://doi.org/10.1016/j.rse.2007.03.012, 2007.

d'Ovidio, F., Monte, S. D., Penna, A. D., Cotté, C., and Guinet, C.: Ecological implications of
eddy retention in the open ocean: a Lagrangian approach, J. Phys. Math. Theor., 46, 254023,
https://doi.org/10.1088/1751-8113/46/25/254023, 2013.

Philip, J. R.: Relation between Eulerian and Lagrangian Statistics, Phys. Fluids Suppl., 10, 69–
71, https://doi.org/10.1063/1.1762507, 1967.

van Sebille, E., Griffies, S. M., Abernathey, R., Adams, T. P., Berloff, P., Biastoch, A., Blanke,
B., Chassignet, E. P., Cheng, Y., Cotter, C. J., Deleersnijder, E., Döös, K., Drake, H. F.,
Drijfhout, S., Gary, S. F., Heemink, A. W., Kjellsson, J., Koszalka, I. M., Lange, M., Lique, C.,
MacGilchrist, G. A., Marsh, R., Mayorga Adame, C. G., McAdam, R., Nencioli, F., Paris, C. B.,
Piggott, M. D., Polton, J. A., Rühs, S., Shah, S. H. A. M., Thomas, M. D., Wang, J., Wolfram, P.
J., Zanna, L., and Zika, J. D.: Lagrangian ocean analysis: Fundamentals and practices, Ocean
Model., 121, 49–75, https://doi.org/10.1016/j.ocemod.2017.11.008, 2018.